# Neural geometry from mixed sensorimotor selectivity for predictive sensorimotor control

**Yiheng Zhang[1,2,3†], Yun Chen[1,2,3†], Tianwei Wang[1,2], He Cui[1,2]***

[1]Center for Excellence in Brain Science and Intelligence Technology, Institute of Neuroscience, Chinese Academy of Sciences, Shanghai, China; [2]Chinese Institute for Brain Research, Beijing, China; [3]University of Chinese Academy of Sciences, Beijing, China

*For correspondence:
hecui@cibr.ac.cn

†These authors contributed equally to this work

## eLife Assessment

This **useful** study examines the neural activity in the motor cortex as a monkey reaches to intercept moving targets, focusing on how tuned single neurons contribute to an interesting overall population geometry. The presented results and analyses are **solid**, though the investigation of this novel task could be strengthened by clarifying the assumptions behind the single neuron analyses, and further analyses of the neural population activity and its relation to different features of behaviour.

**Abstract** Although recent studies suggest that activity in the motor cortex, in addition to generating motor outputs, receives substantial information regarding sensory inputs, it is still unclear how sensory context adjusts the motor commands. Here, we recorded population neural activity in the motor cortex via microelectrode arrays while monkeys performed flexible manual interceptions of moving targets. During this task, which requires predictive sensorimotor control, the activity of most neurons in the motor cortex encoding upcoming movements was influenced by ongoing target motion. Single-trial neural states at the movement onset formed staggered orbital geometries, suggesting that target motion modulates peri-movement activity in an orthogonal manner. This neural geometry was further evaluated with a representational model and recurrent neural networks (RNNs) with task-specific input-output mapping. We propose that the sensorimotor dynamics can be derived from neuronal mixed sensorimotor selectivity and dynamic interaction between modulations.

## Introduction

The motor cortex, a central brain region generating motor commands, is widely known for its relation to movement kinetics (*Evarts, 1968*; *Kalaska et al., 1989*) and kinematics (*Georgopoulos et al., 1982*; *Wang et al., 2022*). More than this, the motor cortex also carries substantial sensory information and is significantly involved in a variety of sensorimotor processes. Many studies have reported that the motor cortex reveals early visuomotor responses for action selection (*Cisek and Kalaska, 2005*; *Wang et al., 2019*) and motor planning (*Hatsopoulos and Amit, 2012*; *Pesaran et al., 2006*; *Rao and Donoghue, 2014*), anticipative activity for complex rules and sequences (*Lu and Ashe, 2005*; *Wang et al., 2024*; *Zimnik and Churchland, 2021*), as well as fast closed-loop modulation for somatosensory and visual feedback control (*Omrani et al., 2016*; *Sobinov and Bensmaia, 2021*; *Stavisky et al., 2017*; *Suway and Schwartz, 2019*; *Tkach et al., 2007*). However, it is still unclear how moving targets requiring forward control affect peri-movement activity in the motor cortex.

Nowadays, the dynamical systems perspective provides a population view to interpret mixed coding for task variables and rules (*Mante et al., 2013*; *Rigotti et al., 2013*). With respect to motor control, the preparatory population activity not only sets initial states to seed the motor generation (*Churchland et al., 2012*; *Churchland and Shenoy, 2024*; *Vyas et al., 2020*), but also presumably contains redundant information – that should be 'held' rather than 'released' to trigger muscles – in an output-null space (*Kaufman et al., 2014*). Therefore, we hypothesize that the potential sensory modulation, which results from target motion and does not intervene the motor output, functions in orthogonal dimensions of output-potent neural states.

To investigate the mixed selectivity and neural dynamics shaped by concurrent sensorimotor signals, we recorded population activity in the primary motor cortex (M1) from monkeys performing a flexible manual interception task. Unlike previous studies constraining interception at a fixed location (*Merchant et al., 2004a*; *Merchant et al., 2004b*), our task demands predictive spatiotemporal mappings to displace a body effector to a trial-varying location. We found that the activity of most neurons was jointly tuned to both reach direction and target motion via directional selectivity shifts, gain modulations, offset adjustments, or their combinations. Strikingly, such mixed sensorimotor selectivity exists throughout the entire trial, in contrast to the gradient of sensory-to-motor tuning from cue to movement epochs in posterior parietal cortex (PPC) that we recently reported (*Li et al., 2022*; *Li et al., 2018*). Principal component analysis (PCA) on the neural population revealed a clear orbital neural geometry in low-dimensional space at the movement onset: The neural states were distributed in reach-direction order and formed ring-like structures whose slopes were determined by target-motion conditions. This target-motion effect is maintained independently of hand speed. Neuronal simulation indicates that these characteristics of neural population dynamics could be derived from the mixed sensorimotor selectivity of single neurons. Recurrent neural networks (RNNs) trained with proper input-output mappings offer insights into the relationship between neuronal modulation and neural geometry in a dynamical system. We propose that this sensory modulation occurs at both single-neuron and population levels as a general element of neural computations for predictive sensorimotor control.

## Results

### Mixed tuning of M1 single neurons during flexible manual interception task

Three monkeys (*Macaca mulatta*, C, G, and D, male, weight 7–10 kg) were trained to perform a delayed manual interception task (*Figure 1A*), which was modified from the task utilized in our recent studies (*Li et al., 2022*; *Li et al., 2018*). To initiate a trial, the monkey needed to hold the center dot of a standing touch screen for 600 ms. Then, a peripheral target appeared, either stationary or rotating around the center dot at a constant angular velocity. The monkey was required to wait during a randomized delay (400–800 ms) until the central dot turned dark (GO signal) and then to immediately reach to the target. Once the monkey touched the screen (Touch) again, the target stopped and another dot showed the touched location, in red for a successful interception or in blue for a failure. The error tolerance between the target and the touched location (reach endpoint) was 3 cm. There were five target-motion conditions, consisting of clockwise (CW) conditions - 240 °/s and −120 °/s, counterclockwise (CCW) conditions 120 °/s and 240 °/s, along with 0 °/s (static) condition. We define the magnitude and direction of the target's velocity as $TV_{mag}$ (e.g. 0 °/s, 120 °/s, and 240 °/s) and $TV_{dir}$ (namely CCW and CW), respectively. These target-motion conditions were interleaved trial by trial, and the initial location of the target was random. The reach endpoints of a well-trained monkey distributed uniformly around the circle (*Figure 1B*, Rayleigh's test: p=0.36; data from monkey C, 772 correct trials in one session). Because the reach direction was defined as the angle of the reach endpoint, for simplicity, we divided the circular space equally into eight sectors (45° per each) and grouped trials according to the eight reach directions and five target-motion conditions.

We recorded neural data with Utah arrays from monkeys C, G, and D (implanted sites are shown in *Figure 1C*, and all datasets are listed in *Supplementary file 1*) and hand trajectories from monkeys C and G (*Figure 1—figure supplement 1*), during the interception task. The hand trajectory was launched to the final interception location. The temporal profiles of hand speed were unimodal

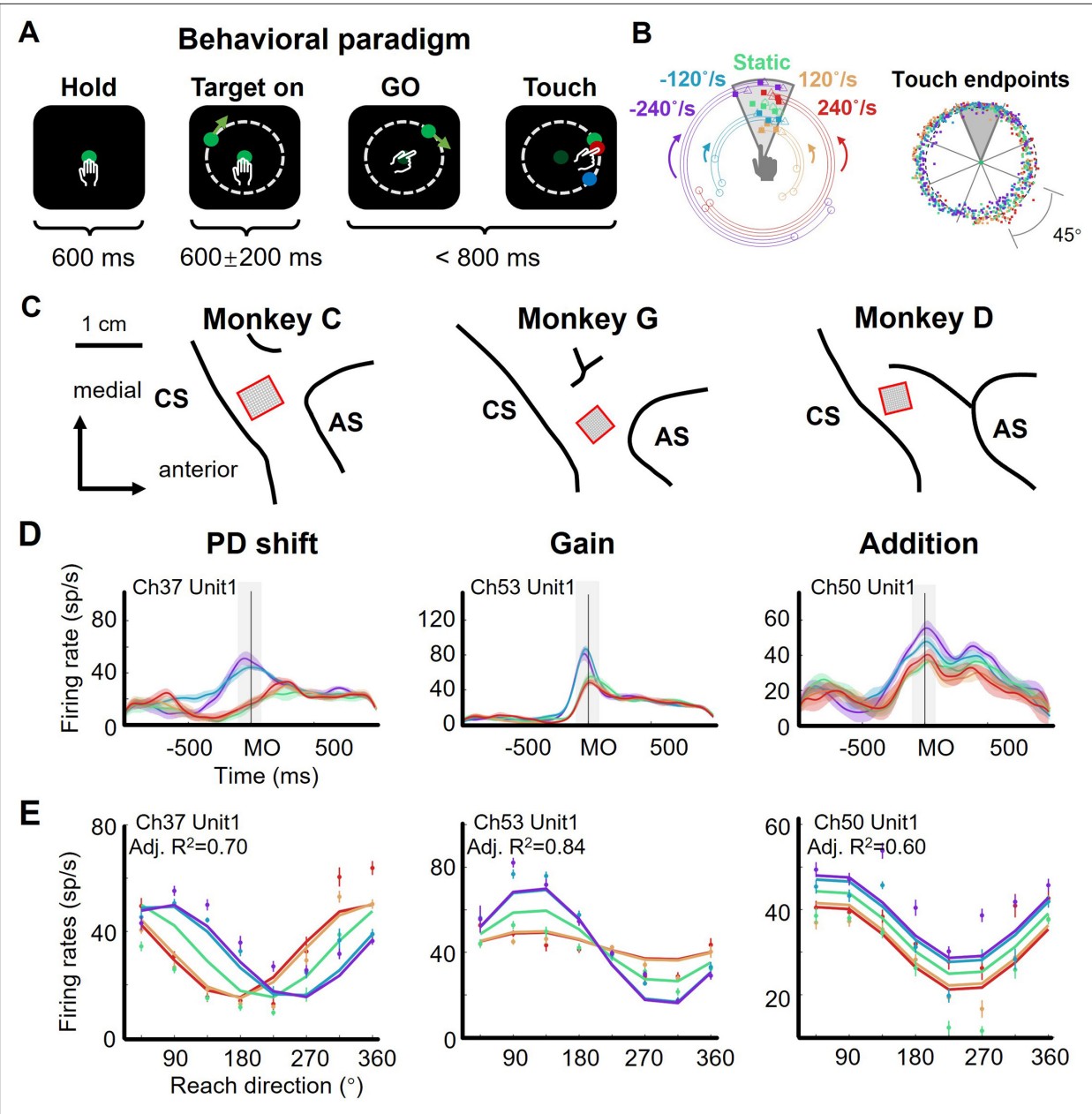

**Figure 1.** The flexible manual interception task and example neurons. (**A**) Task diagram. In each trial, the monkey first holds the center dot for 600ms (Hold), then a target appears (Target on), after 400–800ms delay the center dot turns dark (GO), immediately the monkey moves its hand to reach to the target (Touch). The movement time (from GO to Touch) is required to be within 800ms, otherwise the trial will be aborted. The feedback dot, which is presented at the touched location of the screen, will be in red for success or in blue for failure. (**B**) Distribution of touch endpoints. Left panel shows fifteen reaching-up example trials in five target-motion conditions, three trials in each condition. The squares mark the touch endpoints, while the circles and triangles are the target onset and stop location. The five target-motion conditions (–240°/s, - 120°/s, 0°/s, 120°/s, and 240°/s) are indicated in five colors (purple, blue, green, yellow, and red). Target onset location is randomly distributed. Right panel shows the touch endpoints of all trials, each point represents a trial, colored according to target-motion conditions. The distribution was uniform around the circle (monkey C 772 trials, Rayleigh's test, p=0.36). (**C**) Implanted locations of microelectrode array in the motor cortex of the three well-trained monkeys. Neural data were recorded from the cortical regions contralateral to the used hand. AS, arcuate sulcus; CS, central sulcus. (**D**) Three example neurons with PD shift, gain modulation, and offset addition. The peri-stimulus time histograms (PSTH) show the activity of example neurons when monkeys reached to upper areas in five target-motion conditions. The solid lines represent the trial-averaged firing rates, the colored shadow represents the standard error. The gray shadow indicates the time window between MO-100 ms and MO +100ms. (**E**) The directional tuning curves of the three example neurons with PD shift, gain, and addition modulation around movement onset (MO ±100ms, adjusted R²: 0.70, 0.84, and 0.60). Dots and bars denote the average and standard error of firing rates, colored according to target-motion conditions.

*Figure 1 continued on next page*

*Figure 1 continued*

The online version of this article includes the following figure supplement(s) for figure 1:

**Figure supplement 1.** Flexible manual interception task and behavioral performance.

**Figure supplement 2.** Three example neurons of monkey C.

**Figure supplement 3.** Three example neurons of monkey G.

**Figure supplement 4.** Three example neurons of monkey D.

**Figure supplement 5.** Single-neuron fitting results.

bell-like curves and similar across target-motion conditions (correlation coefficients is 0.97±0.05 for monkey C and 0.99±0.02 for monkey G, mean ± sd).

Notably, we found that neuronal directional tuning during peri-movement period (MO ± 100ms, MO for movement onset) was modulated by target motion, according to our statistical criteria, mainly in three ways: preferred direction (PD) shift, gain modulation, and offset addition (*Table 1*, *Figure 1—figure supplements 2–4*; see Materials and methods). We determined these modulations on the basis of the classical cosine tuning model (*Georgopoulos et al., 1982*) and several previous studies (*Bremner and Andersen, 2012*; *Pesaran et al., 2010*; *Sergio et al., 2005*). Specifically, PD-shift neurons had their PDs shifted in moving-target conditions compared to the static-target condition. As illustrated by the example neuron (*Figure 1D*, *PD shift*), whose PDs corresponding to CCW conditions (red and yellow) and CW conditions (blue and purple) exhibited obvious differences, $TV_{dir}$ rather than $TV_{mag}$ dominated this modulation. Gain-modulation neurons exhibited reach-direction tuning multiplied by target velocity: while their directionality remained invariant, the neuronal responses at PD differed across target-motion conditions. This modulation was dominated by $TV_{dir}$ as well. The turning curves of the example neuron (*Figure 1D*, *Gain*) had higher responses at PD in CW conditions (blue and purple) than in the others (green, yellow, and red), indicating a varying tuning depth for reach direction. Neurons with addition modulation underwent changes of offset activity induced by target velocity (both $TV_{dir}$ and $TV_{mag}$). As shown by the example neuron (*Figure 1D*, *Addition*), this effect was roughly the same in all reach directions.

The activity of these example neurons could be well described by PD shift, gain, and additive models (*Figure 1E*; see Materials and methods), respectively. Nevertheless, it was difficult to classify all neurons with mixed sensorimotor selectivity into one of these three groups exclusively, because many of them experienced a mixture of two or three of above modulations. We found that the adjusted $R^2$ of a full model (0.55±0.24, mean ± sd.) can be higher than that of the PD shift (0.47±0.24), gain (0.46±0.22), additive (0.41±0.26), and simple models (only reach direction, 0.34±0.25) for three monkeys (1162 neurons, rank-sum test, one-tailed, p<0.01, *Figure 1—figure supplement 5*). These target-motion modulations on neuronal directionality suggested the participation of sensory signals in shaping neural dynamics during interception execution.

**Table 1.** Ratio of mixed selectivity neurons around movement onset.

| Units across sessions | Gain (G) | PD shift (S) | Addition (A) | None |
|---|---|---|---|---|
| Monkey C, 7 sessions, N=84.9 ± 15.9 | 46.9 ± 15.9% | 79.0 ± 8.9% | 61.9 ± 14.5% | 6.5 ± 6.2% |
| Monkey G, 4 sessions, N=97.5 ± 17.7 | 29.3 ± 4.9% | 51.6 ± 29.1% | 25.9 ± 8.3% | 25.5 ± 15.8% |
| Monkey D, 4 sessions, N=44.5 ± 7.5, | 45.6 ± 13.8% | 74.4 ± 8.8% | 49.1 ± 9.1% | 10.8 ± 5.4% |
| RNN, N=200, 100 models | 46.5 ± 4.6% | 40.7 ± 4.2% | 57.9 ± 8.9% | 14.2 ± 4.2% |

Single neurons were classified by modulation patterns (more details in Materials and methods). The first column shows the subject, the number of recording sessions, and the number of isolated units (mean ± sd.). Other columns show the ratio of units with any target-velocity modulations in all active units. 'None' means the units were not specifically tuned by either reach direction nor target velocity.

## Coding of sensory and motor information in neural populations

To quantify target motion information embodied in neural response of the motor cortex, we performed a series of decoding analyses on the neural data from monkey C (n=95, 772 correct trials). To begin with, we trained a support vector machine (SVM) classifier for target-motion conditions (chance level: one in five) and another for reach directions (chance level: one in eight; see Materials and methods) on neural data. As *Figure 2A* shows, the decoding accuracy of target-motion condition increased quickly and peaked at over 70% around GO, while the decoding accuracy of reach direction climbed in an approximately linear manner and reached a plateau of about 80% before MO. These results were stable on the data of the other two monkeys and the pseudo-population of all three monkeys (*Figure 2—figure supplement 1*) and reconfirmed by the continuous decoding results with support vector regressions (*Figure 2—figure supplement 2*), suggesting that target motion information existed in M1 throughout almost the entire trial. According to the demixed PCA (dPCA; *Kobak et al., 2016*) results, the reach-direction components occupied high ranks; although the target-velocity components explained few variance, the interaction components were non-negligible (*Figure 2—figure supplement 3*). This implies that the target motion information was intertwined with reach-direction information, rather than being processed independently and in parallel.

Then, we performed another decoding analysis to probe the potential interaction between reach direction and target velocity during execution period. We trained a reach-direction decoder (chance level: one in eight) to check if the decoder of one certain target-motion condition could be transferred to other conditions (*Figure 2B* left). It turned out that the performance of the transferred decoder deteriorated more significantly for $TV_{dir.}$ (CCW *vs*. CW, mean ± sd. of accuracy 0.26±0.06), compared with that for $TV_{mag}$ (120 *vs* 240, 0.50±0.06, paired t-test, p<0.01) and for target's state (static *vs*. motion 0.55±0.06, paired t-test, p<0.01). This result suggests that the coding of reach direction was rather sensitive to $TV_{dir,}$ but contained similarities for static and moving targets. We also compared the neural coding rules across different reach-direction conditions. We trained a target-velocity decoder (chance level: one in five), and similarly checked the transferred decoding accuracy (*Figure 2B* right). We observed that the target-velocity decoder was locked with reach direction, as the transferred decoding accuracy diminished with increasing difference of reach direction. These results qualitatively imply the interaction as that target velocities affected the reach-directional tuning, especially by $TV_{dir.}$ This target-motion effect was most obvious at the MO (*Figure 2—figure supplement 2C*).

To explore how sensory information influences neural dynamics while preserving motor output, we performed PCA on the normalized population activity. We obtained the trial-averaged neural trajectories (five target velocities by eight reach directions, totally 40 conditions) after TO or after GO (*Figure 2—figure supplement 4A*). The distance between neural trajectories grouped by reach-direction conditions was larger than the distance grouped by target-motion conditions, especially after GO (*Figure 2—figure supplement 4B*), consistent with the dPCA results that M1 primarily encoded reach direction rather than target velocity. In addition, the distance of neural trajectories between CCW and CW was much larger than the distance between 120°/s and 240 °/s conditions, indicating that $TV_{dir}$ dominated the target-motion effect, agreeing with the decoding results (*Figure 2—figure supplement 4C*).

However, these neural trajectories were not yet the ideal description, because they were shaped mostly by time. Therefore, to highlight the proposed target-motion effect on reach direction, we focused on four key time windows and snapshotted the neural trajectory as neural state to extract the coding rule at single-trial level and from a geometric view. Here, we define the 'neural states' as the projection of single-trial data during a specific time bin on principal components (PCs; *Mante et al., 2013*; *Parthasarathy et al., 2017*; *Sun et al., 2022*). At the MO, the first two PCs of the neural states explained the most variance ([24.8%, 13.8%]) and were most related to reach direction (the goodness of fitting reach direction, $[R^2_{pc1}, R^2_{pc2}] = [0.82, 0.81]$, *Figure 2C*). While reach direction was represented by the first two PCs at GO and during movement execution, target velocity influenced the tuning pattern of the first three PCs in various ways (*Figure 2D*).

## Orbital neural geometry in latent space

We visualized the neural states in the low-dimensional space spanned by the above three PCs (*Figure 2—figure supplement 2D*). At the MO, the projections of the single-trial neural states onto the PC1-PC2 subspace distributed in reach-direction order, as reach direction was divided into eight

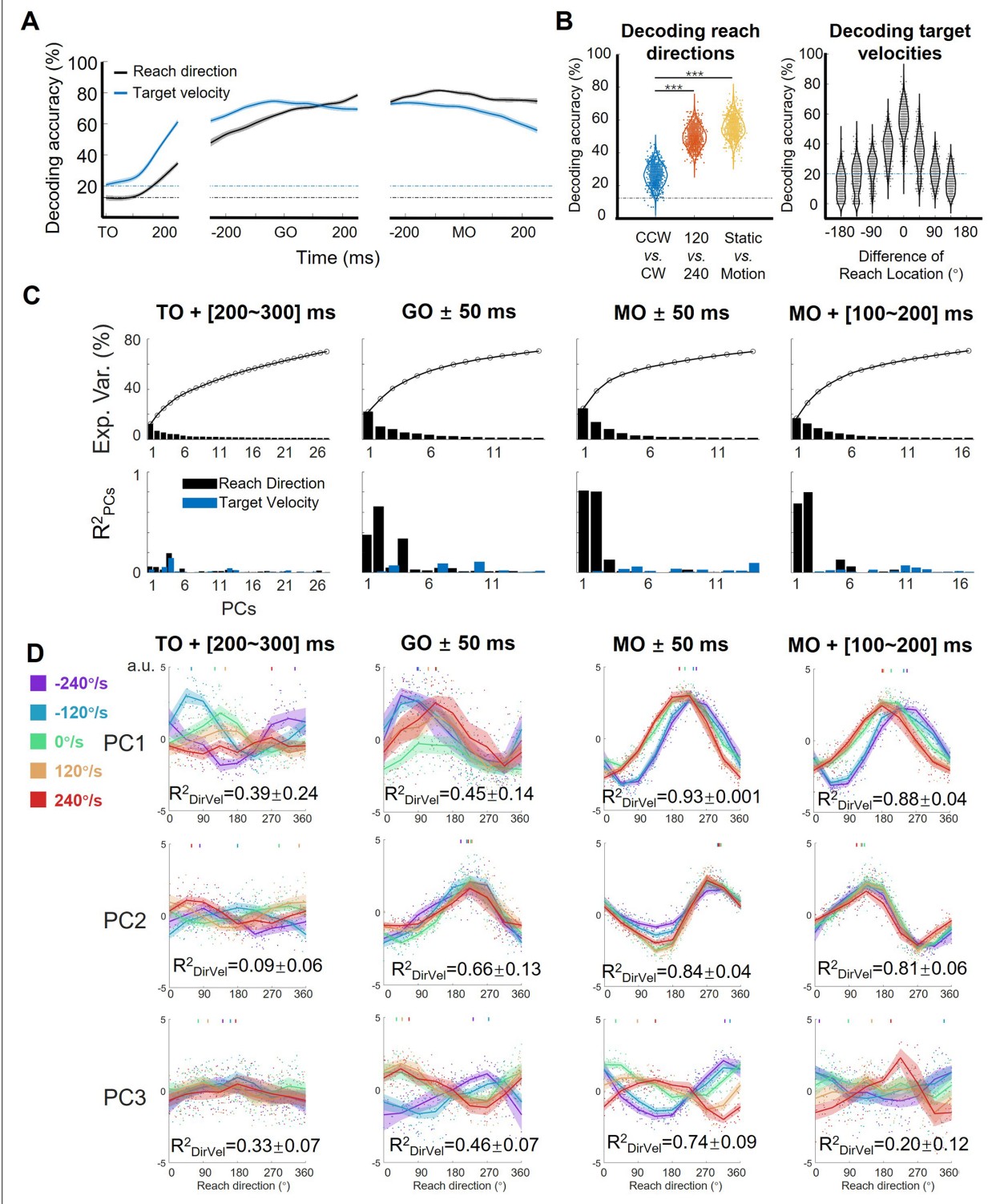

**Figure 2.** Features of the encoding pattern at population level. (**A**) The decoding accuracy (SVM with 10-fold cross-validation) of reach direction (black line) and target velocity (blue line) by population activity (monkey C, n=95, 772 trials), is aligned to target on (TO), GO, and movement onset (MO). The dash-dotted lines are chance level of decoding reach direction (black, one in eight) and target velocity (blue, one in five). The shaded area is the standard deviation of the decoding accuracy for 10 permutations. (**B**) The left panel shows the performance of reach-direction decoder (chance level: one in eight) transferred between different target-motion conditions. The SVM decoder was built on randomly selected 100 trials in training dataset and tested in another 100 trials from a dataset of different conditions (CCW *vs.* CW, 120 *vs* 240, static *vs.* motion). The distributions of decoding accuracy were from 1000 repetitions and compared with one tailed t-test (p<0.01, with three stars). The right panel shows the performance of target-velocity

*Figure 2 continued on next page*

*Figure 2 continued*

decoder (chance level: one in five) in different reach-direction conditions. The accuracy distribution was also obtained from 1000 repetitions. (**C**) The explained variance and representation of the principal components (PCs). The first row shows the explained variance of each PC (cumulatively over 70%). The second row shows the PCs' fitting goodness ($R^2_{PCs}$) of reach direction and target velocity in four epochs. (**D**) Directional tuning curves of the PCs. Each row shows the directional tuning of one PC (the first three PCs in C) in four epochs. Each dot represents a trial, tuning curves are averaged in eight reach directions, and PDs of PCs are indicated by the short lines in the top of subplot by a weighted sum of response. The colors of the lines and dots mean the target-motion conditions, as the same as the legend on the left. The goodness of fitting reach direction ($R^2_{DirVel}$) for the single-trial PCs under single target-motion conditions is shown by mean ± sd. across conditions.

The online version of this article includes the following figure supplement(s) for figure 2:

**Figure supplement 1.** Decoding results of extended datasets.

**Figure supplement 2.** Decoding results and neural states across epochs.

**Figure supplement 3.** dPCA and subspace projection.

**Figure supplement 4.** Neural trajectory during preparatory and peri-movement periods.

sectors, there appeared to be eight reach-direction clusters (*Figure 3A* top and *Figure 3B* left). Interestingly, the neural states under each single target-motion condition formed ring-like structures and the fitted ellipses exhibited concentric shapes (the fitting goodness of ellipses, $R^2 = 0.92 \pm 0.01$, mean ± sd. across target-motion conditions; see Materials and methods). Moreover, these ellipses tilted

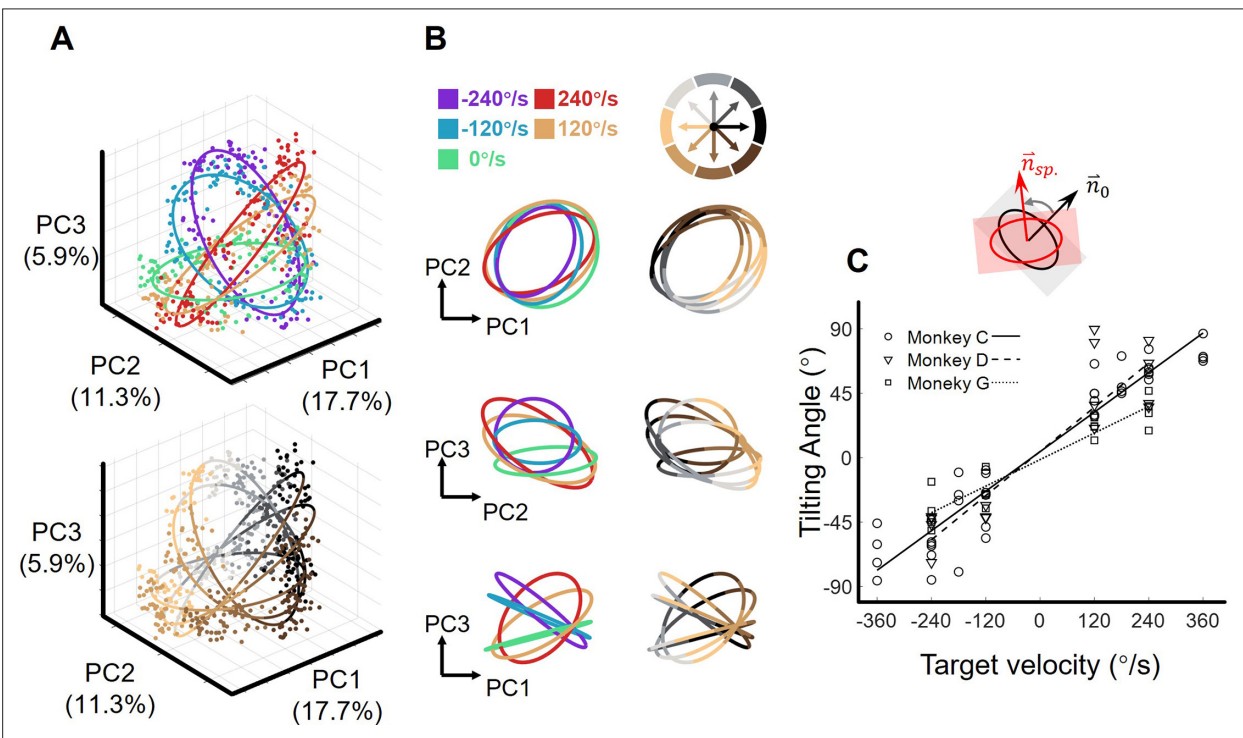

**Figure 3.** The orbital neural geometry in latent dynamics. (**A**) Three-dimensional neural state of M1 population activity obtained by PCA. Each point represents a single trial. The upper subplot is colored according to five target-motion conditions, while the bottom is in colors corresponding to eight reach directions. The explained variances of the first three PCs were 17.7%, 11.3%, and 5.9%. Neural data were from monkey C (N=480, merged six sessions), for each of the total 40 conditions, 15 trials were randomly sampled. (**B**) Fitted ellipses of neural states. The ellipses fitted in (**A**) are projected onto three two-dimensional subspaces, colored by target velocities (left column) or reach directions (right column). (**C**) The relation between the tiling angle and target velocity. The tilting angle is calculated between ellipses of the moving-target conditions and the static-target condition (0 °/s) in the range from −90° to 90°, CCW is defined as positive. Circles, squares, and triangles correspond to monkeys C (7 sessions), G (4 sessions), and D (4 sessions), respectively. The lines indicate the linear fitting between the tilting angle ($\theta$) and target velocity (vel.), with solid line for monkey C ($\theta=0.23$*vel.+4.2, $R^2=0.91$), dashed line for monkey D ($\theta=0.26$*vel.+4.3, $R^2=0.81$), and dotted line for monkey G ($\theta=0.15$*vel.-1.4, $R^2=0.89$).

The online version of this article includes the following figure supplement(s) for figure 3:

**Figure supplement 1.** Neural state of target-motion modulated M1 neurons from three monkeys.

**Figure supplement 2.** Target-motion modulation on hand-speed-filtered trials.

in condition-dependent angles, which is particularly evident in the PC2-PC3 subspace (*Figure 3A* bottom and *Figure 3B* right). Note that here we performed a linear transformation on all resulting neural state points to make the ellipse of the static condition orthogonal to the z-axis for better visualization.

Next, we quantified the spatial features of these ellipses by calculating the tilting angles, which were defined as the angles between the normal vectors of the moving-target and static-target conditions. Strikingly, these tilting angles were linearly correlated with target velocity (both TV$_{mag}$ and TV$_{dir}$), and the relationship was robust in nine datasets from three monkeys (*Figure 3C*, *Figure 3—figure supplement 1*).

To eliminate hand-speed effect, we resampled trials to construct a new dataset with similar distributions of hand speed in each target-motion condition and found similar orbital neural geometry. Moreover, the target-motion gain model provided a better explanation compared to the hand-speed gain model (*Figure 3—figure supplement 2*).

Given these results, we propose that this orbital neural geometry epitomizes the sensorimotor dynamics of M1 at the population level. The sensory input can regularly modulate neural states in an orthogonal dimension (PC3), without interfering with motor generation (in PC1 and PC2).

## Population neural geometry relies on neuronal tuning

To test whether a group of single neurons with a certain type of mixed sensorimotor selectivity could exhibit the orbit neural geometry, three neuronal models were constructed based on the three fitting models described above (PD shift, gain, and addition, see *Figure 1D*, *Figure 1—figure supplements 2–4*). We ran a simulation with these representational neuronal models (*Figure 4A*). Here, each group consisted of 300 model neurons with their PDs uniformly distributed, and was solely modulated as PD-shift, gain, or addition (see Materials and methods). The resulting neural geometry of the three simulation groups showed distinct features (*Figure 4B*): The single-condition ellipses were inclined with target-motion-dependent angles in the PD-shift and gain groups, similar to the real neural data, but the ellipses in the addition group were layered in parallel. The reach-direction clusters in the first two PCs were conservative in the gain and addition group, but not in the PD-shift group. These results indicate that the neural states of the real data mainly resembled the geometry feature of the gain modulation group. Nonetheless, we found that a population with a uniform mixture of all three modulations was able to simulate the neural geometry as well (*Figure 4C*).

Comparing with the static-target condition, we calculated relative rotation angle and tilting angle between ellipses along with the vertical shift of neural states, in order to quantify the simulated structure (*Figure 4D*). The results show that the real data yielded a smaller rotation angle than the PD-shift group, a smaller vertical shift than the additive group, but larger tilting angles than all models. The mixed group had the most similar tilting angle, although with moderate performance in rotation angle and state shift.

These simulations suggest that the existence of PD-shift and additive modulation would not disrupt the neural geometry which is primarily driven by gain modulation; rather, it is possible that these three modulations support each other in a mixed population.

## The recurrent neural network provides dynamic insights

To infer how such modulated subpopulations would interact with each other in a dynamical system, we trained 100 RNN models with random weight initialization (*Figure 5A*; see Methods). The inputs included motor intention, target location, and a GO signal. Motor intention was defined as an abstract motor command predicted to compensate for sensorimotor delays (*Cui, 2016*), and could be provided by the PPC (*Andersen and Buneo, 2002*; *Andersen and Cui, 2009*), here simplified as the interception location. The network was to generate hand velocity after MO. For a fixed validation set of 500 trials, these trained network models performed well (distance between the reach endpoint and the target was 0.0046±0.0027, a. u., mean ± sd., while the radius of target motion was 0.15).

In these network models, we found three comparable features. First, from the decoding result, target motion information existed in nodes' population dynamics shortly after TO (*Figure 5—figure supplement 1A*). Second, most of the activated nodes (for example, *Figure 5B*) could be classified into the above-mentioned three modulations by the same statistical standard as for the real neural data (*Table 1*). Third, the states reduced from node population activity were arranged in a way

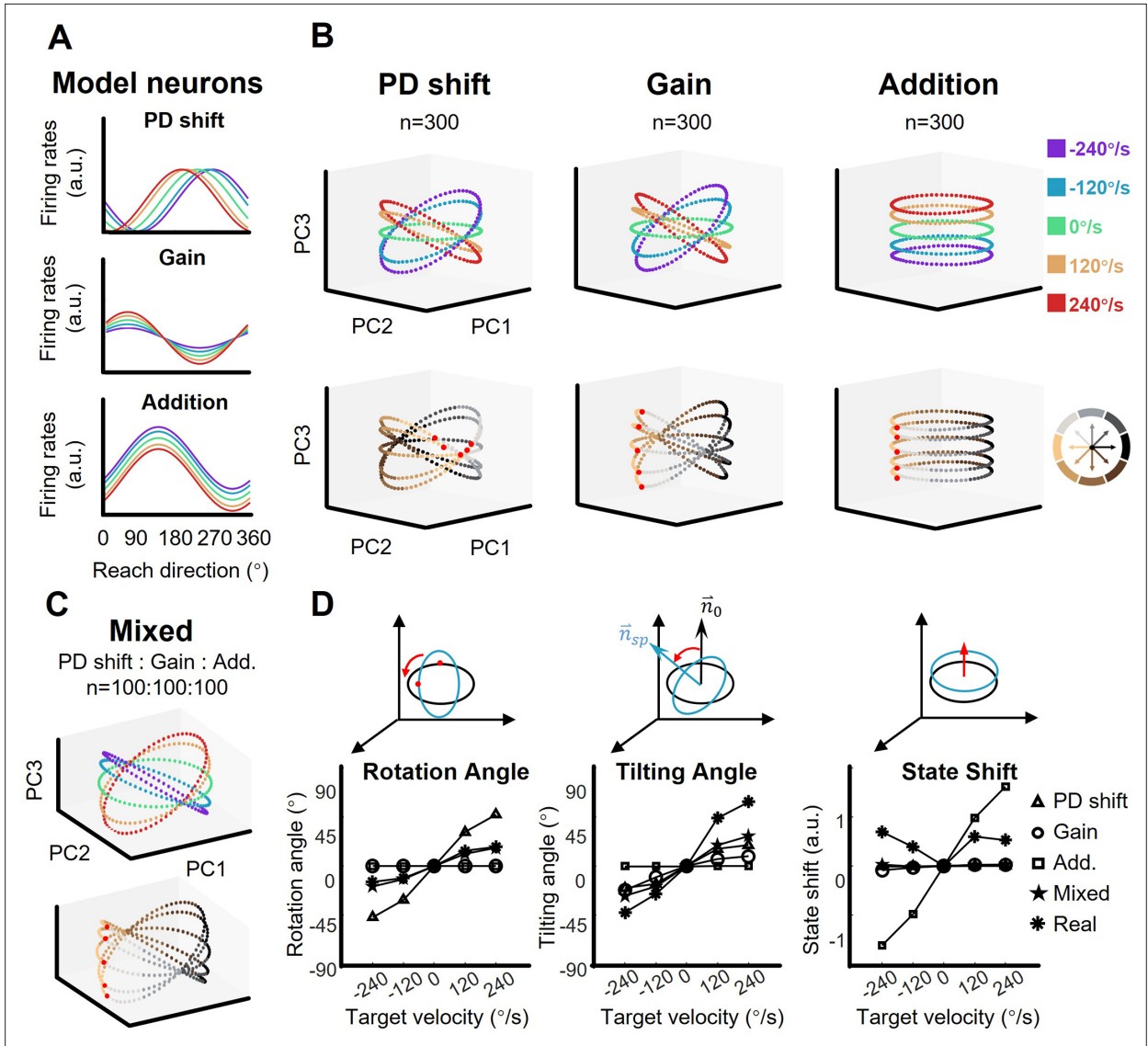

**Figure 4.** The shape of neural dynamics relies on neuronal mixed selectivity. (**A**) The tuning curves of three ideal neurons in five target-motion conditions. From up to down, they are PD shift, gain modulation, and addition. (**B**) The simulated neural states from three groups of ideal neurons, colored according to target-motion conditions (first row) and reach directions (second row). For each simulation, the neural states were obtained from 300 model neurons by PCA. The neural state in 180° reach direction is marked with a red dot. The first two principal components (PCs) can explain more than 95% of the variance in the data (the explained variance of the first three PCs, Gain: 49.5%, 46.6%, and 2.0%; PD shift: 50.1%, 47.1%, and 1.6%; Addition:50.8%, 47.9%, and 1.4%). (**C**) The neural states of a mixed group of 100*3 model neurons, as in (**B**). The explained variance of the first three PCs were 48.4%, 44.2%, and 3.1%. (**D**) Quantification of the difference between neural-state ellipses in four simulated groups and a real dataset (monkey C, n=95). Rotational angle is the angular differences in the first two neural state. Tilting angle is the relative angle of the normal vector of ellipses. State shift is the root of mean squared distance between two ellipses.

resembling the actual neural geometry at the MO (the fitting goodness of ellipses, $R^2$=0.98 ± 0.05, mean ± sd.; *Figure 5C*). The tilting angles followed the same pattern as suggested by the actual results (*Figure 5D*). We then performed canonical component analysis (CCA) and Procrustes analysis (*Supplementary file 2*; see Materials and methods), the results also indicated the similarity between network dynamics and neural dynamics.

With these RNN models, we tried perturbations to investigate the function of certain modulation groups and their connection. The ablation experiments showed that losing any kind of modulation nodes would largely deteriorate the performance, and those nodes merely with PD-shift modulation could mostly impact the neural state structure (*Supplementary file 3*). The connections within

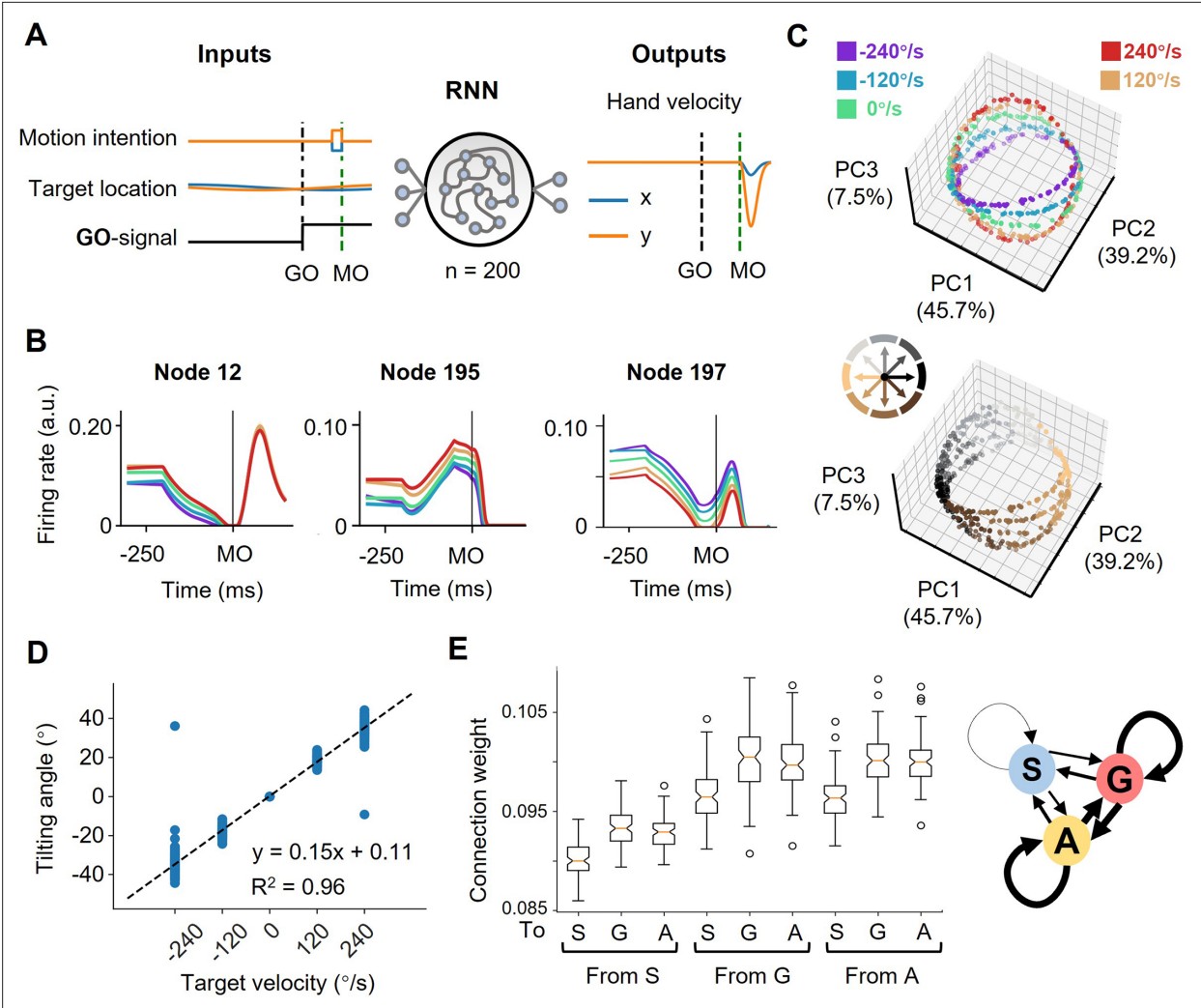

**Figure 5.** The neural geometry in RNNs. (**A**) Network architecture. The network inputs consist of motor intention, target location, and GO signal. The motor intention is the two-dimensional Cartesian coordinate of the interception location, and exists fixed during MO-50ms to MO; the target location is the two-dimensional Cartesian coordinate of the moving target, and appears time-varying during the whole trial; the GO-signal is a step function, jumping from 0 to 1 at GO. The RNN with 200 hidden units is expected to output hand velocity in two-dimensional Cartesian coordinates for accurate interception. (**B**) Three example nodes with PD-shift modulation, gain modulation, and additive modulation. Similar to *Figure 1D*. (**C**) Three-dimensional neural state of node activity obtained by PCA, colored according to target-motion conditions (top) and reach-direction conditions (Bottom). Similar to *Figure 3A*. (**D**) The tilting angle of ellipses. Similar to *Figure 3D*. The fitted line is $\theta$=0.15*vel.+0.11, $R^2$=0.96, across five target velocities. (**E**) The connectivity between different types of modulations. On the left is a boxplot representing the averaged absolute connection weight, across 100 models. S for PD-shift nodes, G for gain nodes, and A for additive nodes. On the right is a diagram of the connectivity, with linewidth representing the relative connection strength.

The online version of this article includes the following figure supplement(s) for figure 5:

**Figure supplement 1.** Decoding results and weight distributions of RNNs.

and between gain modulation nodes and additive modulation nodes were stronger than the others (p<0.01, K-W Test) but comparable to each other (p=1.00, K-W Test; *Figure 5E*). However, further strengthening the connection from gain to additive nodes or within additive nodes would impair the orbital structure more significantly (*Supplementary file 3*). We speculate that the influence, and perhaps the formation, of modulations both relate to the robust input and output weight pattern across models (*Figure 5—figure supplement 1B–C*). From these results, all the three modulations are necessary for anticipative performance and their interactions likely mold the ring-like structure, though their specific roles are not yet elucidative.

We also tested three alternative network models: (1) only receives motor intention and a GO-signal; (2) only receives target location and a GO-signal; (3) initialized with sparse connection (sparsity = 0.1); the unmentioned settings and training strategies were as the same as those for original models (*Supplementary file 4*; see Materials and methods). The results showed that the three modulations could emerge in these models as well, but with obviously distinctive distributions. In (1), the ring-like structure became overlapped rings parallel to the PC1-PC2 plane or barrel-like structure instead; in (2), the target-motion related tilting tendency of the neural states remained, but the projection of the neural states on the PC1-PC2 plane was distorted and the reach-direction clusters dispersed. These implies that both motor intention and target location seem to be needed for the proposed ring-like structure. The initialization of connection weights of the hidden layer can influence the network's performance and neural state structure, even so, the ring-like structure and especially the target-velocity dependent tilting remained in a degree. However, we did not find specific connection patterns in these sparsely-initialized networks (p=0.73, K-W Test).

## Discussion

Mixed selectivity in M1 has been widely reported; but to our knowledge, sensory modulation on anticipatory motor control, as examined in this study, has been scarcely studied. To reveal how sensory context influence neural dynamics during movement, we recorded M1 population activity from monkeys performing a flexible manual interception that is highly dependent on predictive sensorimotor transformations. Single-neuron activity showed that the movement tuning of M1 neurons varied with target-motion conditions in complicated ways, including PD shift, gain modulation, addition, or their mixtures. Unsupervised dimensionality reduction (PCA) on population activity revealed an orbital neural geometry, where neural states of single trials spontaneously formed ellipses and were tilted according to target-motion conditions. Such a neural geometry, which could be simulated with a group of representational neurons for gain modulation alone, also emerged in the RNNs with appropriate input-output mappings. As suggested by the dynamic and connected population of the RNNs, the interactions between modulations were sophisticated in preserving the encoding of motor output. These results reveal sensory modulation of peri-movement neural dynamics at the single-neuron and population levels, and bridge the neuronal mixed sensorimotor selectivity and a low-dimensional neural geometry.

A main concern about our finding is that the observed modulation would actually be attributed to interactions between kinematic variables, such as reach direction and hand speed (*Inoue et al., 2018*; *Moran and Schwartz, 1999*; *Paninski et al., 2004*). However, we can eliminate the hand-speed effect for at least two reasons: First, the hand speed pattern and neural activity pattern were not consistent. We found hand speed vary with $TV_{mag}$, but the largest gap in neuronal activity related to $TV_{dir}$ (*Figure 1*, *Figure 1—figure supplement 1*); Second, the neural geometry was not contingent upon hand speed distribution, as similar hand-speed distribution across target-motion conditions did not change the phenomenon (*Figure 3—figure supplement 2*). It is a pity that we did not record sufficient muscle data during interception to rule out the motor cortical representation for kinetic variables such as force (*Evarts, 1968*; *Scott and Kalaska, 1997*; *Sun et al., 2022*), but we believe that this would not invalidate our main conclusions.

Another concern is whether the neural geometry was representing sensory information. Previous studies have showed that M1 has transient responses to behaviorally relevant visual cues (*Alexander and Crutcher, 1990*; *Lamarre et al., 1983*; *Lurito et al., 1991*; *Port et al., 2001*; *Rao and Donoghue, 2014*; *Rizzolatti et al., 1981*), but it is seldom evoked by purely visual stimuli, only weakly tuned (*Kruse et al., 2002*; *Kwan et al., 1985*; *Merchant et al., 2001*). Here, the modulated neural dynamics, though distinguished according to target-motion conditions, are ultimately shaped for motor generation rather than sensory representation. To further clarify, the discussing target-motion effect is different from the sensory modulation in action selection (*Cisek and Kalaska, 2005*), motor planning (*Pesaran et al., 2006*), visual replay and somatosensory feedback (*Pruszynski et al., 2011*; *Stavisky et al., 2017*; *Suway and Schwartz, 2019*; *Tkach et al., 2007*), because it occurred around movement onset and in predictive control trial-by-trial.

In the present study, we focused on the orbital neural geometry by taking 'snapshots' of neural activities with PCA. In addition to this unsupervised dimensionality reduction method, we also tried another supervised one – dPCA (*Kobak et al., 2016*). However, the dPCA results (*Figure 2—figure*

*supplement 3*) showed that the condition-independent temporal component explained the largest variance, followed by the component of reach direction, whereas the interactive and target-velocity components only covered very small proportions. The dPCA did not distinguish an independent subspace coding target-velocity (*Figure 2—figure supplement 3C*), suggesting the target-velocity information persisted in the shared subspace with reach direction.

It would be interesting to explore whether other motor areas also allow sensory modulation during flexible interception. The functional differences between M1 and other areas lead to uncertain speculations. Although M1 has pre-movement activity, it is more related to task variables and motor outputs. Recently, a cycling task sets a good example that the supplementary motor area (SMA) encodes context information and the entire movement (*Russo et al., 2020*), while M1 preferably relates to cycling velocity (*Saxena et al., 2022*). The dorsal premotor area (PMd) has been reported to capture potential action selection and task probability, while M1 not (*Cisek and Kalaska, 2005*; *Glaser et al., 2018*; *Wang et al., 2019*). If the neural dynamics of other frontal motor areas are revealed, we might be able to tell whether the orbital neural geometry of mixed selectivity is unique in M1, or it is just inherited from upstream areas like PMd. Either outcome would provide us some insights into understanding the interaction between M1 and other frontal motor areas in motor planning.

From a bigger view, how could the motor cortex cooperate with other brain regions during flexible interception? It could be true that the communication and collaboration of PPC-motor cortex circuity translate abstract motor intention to executable motor command: PPC transforms sensory modulation to the motor cortex, while the motor cortex sends potential movement direction to PPC, and then their recurrent interaction generates mature motor intention and command. Our simulations with neuronal models and RNNs demonstrate that differences in modulations and the interaction between them can be essential. In fact, modulations in the form of PD shift and gain have been widely found in the motor cortex and PPC. For example, motor cortex neurons experience gain and PD shift modulation by arm posture (*Kakei et al., 2001*; *Kakei et al., 1999*; *Scott and Kalaska, 1997*). Neurons in PPC have eye and hand gain fields for a visually-guided reach plan (*Batista et al., 1999*; *Buneo et al., 2002*; *Chang et al., 2009*), integrating target position related to gaze and hand position to form motor intention (*Andersen et al., 1997*; *Andersen and Buneo, 2002*; *Cohen and Andersen, 2002*). Therefore, we posit that the three types of modulations should all be involved in sensorimotor computation in predictive motor control. Furthermore, recent studies show that interactions of PPC-motor cortex circuity are involved in motor planning, spatial transformation and motor selection (*Dann et al., 2016*; *Martínez-Vázquez and Gail, 2018*; *Schaffelhofer and Scherberger, 2016*). The PPC-M1 circuit, as a key part of cortico-subcortical networks for the predictive sensorimotor control (*Brozović et al., 2007*), will be a topic for future studies.

# Materials and methods
## Experimental model and subject details

All procedures have been approved by the Biomedical Research Ethics Committee of Shanghai Institutes for Biological Sciences, Chinese Academy of Sciences, under permit number ER-SlBS-221603P. They comply with national and local laws and regulations in China and are in accordance with the Guide for Care and Use of Laboratory Animals of the Institute for Laboratory Animal Research (version 20160310). All surgery was performed under anesthesia, and every effort was made to minimize suffering. The details of the experimental procedures are as follows. Three adult male rhesus macaques (monkey C, D, and G, *Macaca mulatta*, 7–10 kg) were recruited in this study.

## Task and behavior

The monkeys sat in primate chairs to perform the task. The stimuli were back-projected onto a vertical touch screen (Elo Touch system; sampling at 100 Hz, spatial resolution <0.1 mm) about 30 cm in front of the monkeys. The monkeys were trained to perform a flexible manual interception task in a dark room. The task paradigm was modified based on the visually guided reaching interception task in a previous study (*Li et al., 2018*). In the beginning, the monkey held the green central dot of a touch screen for 600ms to initiate a trial (*Figure 1A*). Then, a green target dot appeared at a random location, moving along a circular path with a radius of 15 cm around the central dot. The central dot turned dark as a GO cue after a random delay (400–800ms); then the monkey could intercept

the target at any moment within 150–800ms after the GO cue. Once any peripheral location was touched, the target stopped. The tolerance range of the touch endpoint for correct trials was within 3 cm of the target. The monkey would be rewarded with juice after each correct trial. Conditions where targets moved clockwise (CW; –240 °/s, –120 °/s) or counterclockwise (CCW; 120 °/s, 240 °/s), as well as targets stayed stationary (0°/s), were pseudo-randomly interleaved trial by trial. Additional target velocities (–360 °/s, –180 °/s, 180 °/s, 360 °/s added) were introduced in subsequent sessions (*Supplementary file 1*). The single-trial MO is defined as the moment when hand velocity first rose to 5% of the maximum. Task control and behavior data acquisition were managed via MonkeyLogic v2.0 [https://monkeylogic.nimh.nih.gov/index.html] (*Hwang et al., 2019*).

## Data collection

After the monkeys were adequately trained for the task (successful rate >90%), head-posts were implanted stereotaxically under anesthesia (induced by 10 mg/kg ketamine, then sustained by 2% Isoflurane). After a few weeks of recovery and adaptation, the monkeys were implanted with Utah microelectrode arrays (Blackrock Microsystems, Salt Lake City, UT) in M1 of the hemisphere contra-lateral to the handedness (*Figure 1C and a* 128-channel array for monkey C, 96-channel arrays for monkey G and D). The recording areas were identified by Magnetic Resonance Imaging (MRI) and cortical sulcus features. Neuronal activity was recorded via a Cerebus acquisition system (256-channel recording system Blackrock Microsystems), sampled at 30 kHz. We collected 85±16, 45±8, 98±18 well-isolated units from monkey C, G, and D across sessions, respectively (mean ± sd., more details in *Supplementary file 1*). Array-recorded raw data were sorted offline by Wave_clus (*Quiroga et al., 2004*). Hand trajectory was collected by optical camera (VICON Inc 100 Hz) with an infrared marker on the fingertip from GO to Touch, and touch endpoint was collected by touchscreen.

## Peri-stimulus time histograms (PSTHs)

The PSTHs and spike rasters of single neurons are shown in *Figure 1D*, *Figure 1—figure supplements 2–4*. All trials were classified into 40 conditions, eight reach-endpoint sectors by five target-motion conditions. Condition-averaged firing rates were calculated with 50 ms bins and smoothed with a Gaussian kernel (standard deviation = 20ms). The standard error of firing rates was estimated from the 10 bootstrap samples in the trials of corresponding condition.

## Classification of neuronal tuning properties

To depict target-motion modulation for single-neuron reach-direction tuning, we applied a series of statistical analyses for classification.

We first calculated three indices of each neuron for each target-motion conditions, using the trial-averaged firing rate around movement onset (MO ± 100ms). These indices were preferred direction (PD), tuning depth, and offset activity. The PD of each neuron for a certain target-motion condition was calculated by the weighted sum of neuronal firing rates averaged by eight reach-direction sectors. The tuning depth of each neuron was determined by the range (max - min) of firing rates in corresponding target-motion conditions. The offset activity of each neuron was calculated by the mean firing rate (MO ± 100ms) of each target-motion condition.

Then, we classified PD shift, gain, and addition groups (in *Table 1*). A neuron was classified as 'PD shift', if its PDs were significantly different between the moving-target conditions and the static-target condition Watson-Williams test in circular data, CircStat by *Berens, 2009*; as 'gain', if its tuning depths were significantly different between the moving-target conditions and the static-target condition (two-tailed Wilcoxon signed-rank test, $p<0.05$); as 'offset', if its offset activities were significantly different between the moving-target conditions and the static-target condition (two-tailed Wilcoxon signed-rank test, $p<0.05$).

## Population decoding

The population activity of the motor cortex was used to decode target motion and reach direction by SVM. Neuronal firing rate was soft-normalized as

$$FR_{norm.} = \frac{FR_{raw}}{FR_{max} - FR_{min} + 5}$$

where raw firing rates were divided by the range of firing rates plus five (*Churchland et al., 2012*). We trained two SVM classifiers (MATLAB function 'fitcecoc', 10-fold cross-validation) to decode reach direction (chance level: one in eight) and target motion (chance level: one in five) of single trials in 100 ms window sliding with 50 ms step (*Figure 2A*). The temporal decoding was repeated ten times to obtain the mean and standard deviation of decoding accuracy.

We tested the generalization of reach-direction and target-motion decoders (SVM, MATLAB function 'fitcecoc') in different conditions during execution period (MO ± 100ms, *Figure 2B*). The decoder predicted single-trial reach direction (one of the eight 45° sectors) or target motion (one of the five ones) across conditions based on normalized population activity. The reach-direction decoder (*Figure 2B* left), which was trained by trials in a certain set of target-motion conditions, was tested with trials from another set of target-motion conditions (CCW vs. CW, or 120 vs 240, or static vs. motion, 100 trials randomly selected without replacement from corresponding training-test datasets), and the training-test decoding was repeated 1000 times to compare the distribution of accuracy with paired t-test. The target-velocity decoder (*Figure 2B* right), which was trained by trials in a given reach-direction condition within 45°, was tested with trials from other reach-direction conditions.

## Unsupervised dimensionality reduction

The population activity was stored in NKT datasets, where N, K, and T denote the neurons, trials, and time bins, respectively. For neural state, we averaged T dimension of neural activity in a 100 ms bin (for example, the two 50 ms bins around MO), and normalized neuronal firing rates by Z-score (MATLAB function 'zscore') to get a K x N dataset. After preprocessing, we used PCA (MATLAB function 'pca') to reduce the dimension from K x N to K x P (P is the number of PCs), and we fitted the PCs with reach direction ($PC = a_1 * cos(\theta) + a_2 * sin(\theta) + c$) and target velocity ($PC = \frac{a_1}{1 + e^{-a_2(vel.)}} + c$) in *Figure 2*. We also tried independent component analysis (ICA, MATLAB function 'rica'), with similar results in *Figure 2—figure supplement 2B*. Neural states of single trials were colored according to target velocity or reach direction and fitted as three-dimensional (first three PCs, *Figure 3*) ellipses by MATLAB package 'MatGeom' (*Legland, 2025*). To show the relative position of ellipses in the best viewing angle, we used an isometric affine transformation to globally map all neural state points on new axes, while preserving the proportional relationships between points. After this linear transformation, the azimuth and elevation of ellipses changed slightly, but the tilting angle between ellipses remained constant (*Figure 3*). The tilting angles, rotation angles, and state shift were calculated between ellipses of the moving-target conditions and the static-target condition in each session, with CW defined as negative angles and CCW as positive angles. A set of tilting angles were obtained from corresponding conditions in one dataset, and a linear regression model was used to fit all ellipses angles (θ) and target velocities (vel.) in nine sessions. For neural trajectory, we averaged NKT to NCT (40 condition, five target velocities by eight reach directions) and reduced neural dimension N to explain concatenated C x T (40 conditions * 10 bins) with PCA. The neural trajectories of preparatory and peri-movement period (TO~+500ms, GO~+500ms) were shown in *Figure 2—figure supplement 4* with Euclidian distance across conditions.

## Fitting and simulating single-neuron activity

We used PD shift, gain, offset, and full models to fit neuronal activity. Neurons are fitted by single-trial data. We introduced a special sigmoid function to fit the nonlinear target-motion effects because target-velocity direction (CCW vs. CW) has a stronger effect than target-velocity magnitude (120 vs 240; *Churchland and Lisberger, 2001*; *Pouget and Sejnowski, 1997*). The gain and additive models refer to hand-velocity gain studies (*Amirikian and Georgopoulos, 2000*; *Inoue et al., 2018*; *Moran and Schwartz, 1999*).

In the gain model, the target-motion effect on the amplitude of cosine tuning is denoted as:

$$FR = \left( \frac{a_1}{1 + e^{-a_2(vel.)}} + c_2 \right) * cos(\theta - \theta_{pd}) + c_1$$

where $FR$ is the firing rate at the movement onset (MO ± 100ms). $\theta$ and *vel.* are the reach direction and target velocity, respectively; $\theta_{pd}$ is the fitted preferred direction of the neuron; $a_1, a_2, c_1$ are constants to be fitted.

In the additive model, the target velocity adjusts the offset activity, as:

$$FR = a_1 * cos\left(\theta - \theta_{pd}\right) + \frac{a_3}{1 + e^{-a_2\left(vel.\right)}} + c_1$$

with similar symbols to the gain models, plus the new constant $a_3$.

In the PD shift model, the target-motion effect on PDs is represented as:

$$FR = a_1 * cos\left(\theta - \theta_{pd} + \frac{a_3}{1 + e^{-a_2\left(vel.\right)}}\right) + c_1$$

with the similar symbols to the above models.

The full model integrates all the three of the above effects:

$$FR = a_1 * cos\left(\theta - \theta_{pd}\right) + \frac{a_2}{1 + e^{-a_2\left(vel.\right)}} + \frac{a_3}{1 + e^{-a_2\left(vel.\right)}} * cos\left(\theta - \theta_{pd}\right) + a_4 *$$

$$cos\left(\theta - \theta_{pd} - \frac{a_5}{1 + e^{-a_2\left(vel.\right)}}\right) + c_1$$

with constants $a_1, a_2, a_3, a_4, a_5$.

We fitted neuronal activity with these four models (MATLAB 'fit' function), and compared the fitting goodness with adjusted R-squares ($R_{adj.}^2 = \frac{(1-r^2)(n-1)}{n-p-1}$, where $r$ is the original R-square, $n$ is the trial number, and $p$ is the degree of the polynomial).

Simulations with model neurons were based on three models to investigate the relationship between neuronal tuning (**Figure 4A**) and population neural geometry (**Figure 4B**). We first built three model neuron groups (each n=300) and performed PCA to obtain the neural state. Then, we repeated this in a mixed group, including 100 neurons in each of PD shift, gain and additive model (**Figure 4C**). Model neurons had three components: cosine-tuning for reach direction ($\theta_{pd,n}$, PD uniformly distributed around the circle for each neuron $n$), Gaussian temporal profiles ($t$=1:200, σ=30, peak time $t_{\mu,n} \in N\left(100, 10\right)$, the 100-th bin is the MO, random distribution for $t_{\mu,n}$ of neurons), and distinct target-motion modulation for each group. We designed five target velocity values ($vel. = \left[1, 2, 3, 4, 5\right]$) and 64 reach directions ($\theta = \frac{360*\left[1:64\right]}{64}$), for 320 trials in total (**Michaels et al., 2016**). The concrete expression of model neurons follows as below:

The gain-model neuron:

$$FR_{t,n,vel.,\theta} = e^{\frac{-\left(t - t_{\mu,n}\right)^2}{2 * 30^2}} * \left(\frac{1}{1 + e^{-g_n\left(vel.-3\right)}} * cos\left(\theta - \theta_{pd,n}\right) + 1\right)$$

here, $g_n$ is randomized within [0, 1] for different neurons, modulating the target-motion gain in a sigmoidal function.

The PD-shift model neuron:

$$FR_{t,n,vel.,\theta} = e^{\frac{-\left(t - t_{\mu,n}\right)^2}{2 * 30^2}} * \left(cos\left(\theta - \theta_{pd,n} - \frac{90}{1 + e^{-s_n\left(vel.-3\right)}}\right) + 1\right)$$

here, $s_n$ is a random value within [0, 1.5] for different neurons, and contributes to a sigmoidal function with target-motion shift on neuronal PD ($\theta_{pd,n}$).

The additive model:

$$FR_{t,n,vel.,\theta} = e^{\frac{-\left(t - t_{\mu,n}\right)^2}{2 * 30^2}} * \left(cos\left(\theta - \theta_{pd,n}\right) + \frac{1}{1 + e^{-g_n\left(vel.-3\right)}} + 1\right)$$

here, $g_n$ is also a random value within [0, 1] for different neurons, and adjusts the offset activity in a sigmoidal function with target-motion.

The firing rates of three groups of model neurons (K x N x T, 320x300 x 200) were averaged at the MO (mean T = [50:150]) to get the K x N (320x200) dataset for PCA. As with the real neural data, we selected the first three PCs (K x C, 320x3) to derive the simulated neural state shown in **Figure 4**.

## RNNs training

For the inputs of RNNs, motor intention appears from MO-50 ms to MO, and is represented as constant variables in the form of two-dimensional Cartesian coordinates; target location is designed as time-varying two-dimensional Cartesian coordinates of the target throughout the entire trial; the GO-signal is a step function jumping from 0 to 1 at GO. Each RNN consists of 200 hidden units, and outputs hand velocity for accurate interception after the MO.

The RNN nodes evolve according to a standard dynamic differential equation:

$$\tau \dot{x} = -x + Jr + Bu$$

where $\tau$ is a time constant (here 50 ms), $x$ is the activity, $r$ is the firing rate, and $u$ denotes the combined inputs. $r$ can be calculated from $x$ following:

$$r = \begin{cases} tanh\left(x\right), x > 0 \\ 0, x \leq 0 \end{cases}$$

The connection matrix $J$ of hidden layer was randomly initialized with a normal distribution (mean = 0, sigma = $g/\sqrt{N}$, g=1.5, N=200), and the connection between inputs and hidden units, the matrix $B$, was initialized as all zero. The initial states were zero vectors. The output $z$ is obtained by

$$z = Wr$$

where $W$ is the read-out weight, and is expected to reproduce the desired hand velocity generated by bell-shaped physical equation (*Kao et al., 2021*). $W$ was also initiated as zero matrix. The loss function is:

$$E = e + \alpha r_1$$

where $e$ is the mean squared error of $z$ and training target. $r_1$ is a regularity (*Sussillo et al., 2015*), denoting the magnitude of the nodes' activity and calculated as activity squared summed across time. $\alpha = 1e - 7$. The optimization was realized with optim.Adam() based on PyTorch, the learning rate was 0.001.

## RNNs analyses

The classification of modulation type on network nodes was finished with the same procedure for real data as above. The firing rates of nodes ([0, 1]) were used for all analyses.

For decoding, similar to *Population decoding*, two SVM classifiers (Python, SVC from sklearn.svm, 10-fold cross-validation) were trained to decode reach direction and target motion of single trials in 100 ms window sliding with 50 ms step. It should be noted that, instead of original firing rates, the first 100 PCs were used for decoding, in order to decrease the influence of inactivated nodes. The decoding results were first averaged cross 10 cross-validation repetitions and then the resulting means were averaged across 100 models to obtain mean ± sd.

We performed Canonical Component Analysis (CCA) and Procrustes analysis to validate the similarity between data and network dynamics. The data were around the TO or around the MO each in 2 seconds (±1 s, 50 ms x 40 bins). The RNN activity were collected between MO-200 ms and MO +120ms, considering the shortest movement time, and then averaged into 40 bins for comparison. First, we performed PCA (Python, PCA from sklearn.decomposition) on actual and model data to get the first 30 PCs (KT x N → KT x 30), respectively. Then, we used these two-dimensional matrices to compute their first ten canonical components with CCA (Python, CCA from sklearn.cross_decomposition). The Pearson correlation coefficients (Python, pearsonr from scipy.stats) were calculated between paired canonical components. The Procrustes disparity (Python, procrustes from scipy.spatial) were computed between pairs of data and network first 30 PCs.

## RNNs experiments

We performed ablation of nodes and manipulation of connection weights. In former, we selected the nodes from a certain type of modulation, and set all its connection weights to zero to simulate 'ablation'. When manipulating the connection between certain types of modulations, the selected connection weight was adjusted to be 1.5 times the original. Both above perturbations were retained

for the entire trial. The perturbed network was tested with the same validation set of 500 trials as for the intact network.

We also constructed three alternative models to check the influence of input and weight initialization. The first one only receives motor intention and a GO-signal (GM); the second only receives target location and a Go-signal (GT); these two were initialized and trained as the main model. The third one was initialized with sparse connection (sparsity = 0.1); except for this, the input and the training were as the same as the main model. All involved modes shared the same validation set.

## Software

We used the Python 3.11 (python.org) and MATLAB 2018b (The MathWorks, Inc).

## Acknowledgements

We thank C Li, J Malpeli, C Zheng, and R Zheng for helpful comments and discussions; C Guan for veterinary assistance; and P Ding, L Du, and Z Xiao for administrative support.

## Additional information

### Funding

| Funder | Grant reference number | Author |
| --- | --- | --- |
| National Key Research and Development Program of China | 2020YFB1313400 | He Cui |
| National Key Research and Development Program of China | 2017YFA0701102 | He Cui |
| National Natural Science Foundation of China | 31871047 | He Cui |
| National Natural Science Foundation of China | 31671075 | He Cui |
| Chinese Academy of Sciences | Strategic Priority Research Program of Chinese Academy of Sciences XDB32040103 | He Cui |

The funders had no role in study design, data collection and interpretation, or the decision to submit the work for publication.

### Author contributions

Yiheng Zhang, Conceptualization, Data curation, Formal analysis, Investigation, Methodology, Writing – original draft, Writing – review and editing; Yun Chen, Conceptualization, Formal analysis, Investigation, Methodology, Writing – original draft, Writing – review and editing; Tianwei Wang, Conceptualization, Data curation, Investigation, Methodology, Writing – original draft, Writing – review and editing; He Cui, Conceptualization, Resources, Supervision, Funding acquisition, Investigation, Methodology, Writing – original draft, Project administration, Writing – review and editing

### Author ORCIDs

Yiheng Zhang ⓘ https://orcid.org/0000-0002-5370-1316
Yun Chen ⓘ https://orcid.org/0000-0002-0817-2160
Tianwei Wang ⓘ https://orcid.org/0000-0002-5192-5594
He Cui ⓘ https://orcid.org/0000-0001-6277-9804

### Ethics

All procedures have been approved by the Biomedical Research Ethics Committee of Shanghai Institutes for Biological Sciences, Chinese Academy of Sciences, under permit number ER-SIBS-221603P. They comply with national and local laws and regulations in China and are in accordance with the

Guide for Care and Use of Laboratory Animals of the Institute for Laboratory Animal Research (version 20160310). All surgery was performed under anesthesia, and every effort was made to minimize suffering.

Reviewer #1 (Public review): https://doi.org/10.7554/eLife.100064.3.sa1
Reviewer #2 (Public review): https://doi.org/10.7554/eLife.100064.3.sa2
Reviewer #3 (Public review): https://doi.org/10.7554/eLife.100064.3.sa3
Author response https://doi.org/10.7554/eLife.100064.3.sa4

## Additional files

### Supplementary files
Supplementary file 1. Neural datasets.

Supplementary file 2. Neural dynamics similarity.

Supplementary file 3. RNN perturbation results.

Supplementary file 4. Alternative models.

MDAR checklist

### Data availability
The example experimental datasets and relevant analysis code have been deposited in Mendeley Data. The RNN relevant code and example model datasets are available on GitHub (copy archived at *Chen, 2025*).

The following dataset was generated:

| Author(s) | Year | Dataset title | Dataset URL | Database and Identifier |
| --- | --- | --- | --- | --- |
| Zhang Y, Chen Y, Wang T, Cui H | 2025 | Neural Geometry from Mixed Sensorimotor Selectivity for Predictive Sensorimotor Control | http://doi.org/10.17632/8gngr6tphf.2 | Mendeley Data, 10.17632/8gngr6tphf.2 |

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
