## [Editor Report · eLife Assessment]

This **useful** study examines the neural activity in the motor cortex as a monkey reaches to intercept moving targets, focusing on how tuned single neurons contribute to an interesting overall population geometry. The presented results and analyses are **solid**, though the investigation of this novel task could be strengthened by clarifying the assumptions behind the single neuron analyses, and further analyses of the neural population activity and its relation to different features of behaviour.

---

## [Referee Report · Reviewer #1 (Public review)]

Summary:

This study addresses the question of how task-relevant sensory information affects activity in motor cortex. The authors use various approaches to address this question, looking at single units and population activity. They find that there are three subtypes of modulation by sensory information at the single unit level. Population analyses reveal that sensory information affects the neural activity orthogonally to motor output. The authors then compare both single unit and population activity to computational models to investigate how encoding of sensory information at the single-unit level is coordinated in a network. They find that an RNN that displays similar orbital dynamics and sensory modulation to motor cortex also contains nodes that are modulated similarly to the three subtypes identified by the single unit analysis.

Strengths:

The strengths of this study lie in the population analyses and the approach of comparing single-unit encoding to population dynamics. In particular, the analysis in Figure 3 is very elegant and informative about the effect of sensory information on motor cortical activity. The task is also well designed to suit the questions being asked and well controlled.

It is commendable that the authors compare single-unit to population modulation. The addition of the RNN model and perturbations strengthen the conclusion that the subtypes of individual units all contribute to the population dynamics.

Weaknesses:

The main weaknesses of the study lie in the categorization of the single units into PD shift, gain and addition types. The single units exhibit clear mixed selectivity, as the authors highlight. Therefore, the subsequent analyses looking only at the individual classes in the RNN are a little limited. Another weakness of the paper is that the choice of windows for analyses is not properly justified and the dependence of the results on the time windows chosen for single unit analyses is not assessed. This is particularly pertinent because tuning curves are known to rotate during movements (Sergio et al. 2005 Journal of Neurophysiology).

This study uses insights from single-unit analysis to inform mechanistic models of these population dynamics, which is a powerful approach, but is dependent on the validity of the single-cell analysis, which I have expanded on below.

I have clarified some of the areas that would benefit from further analysis below:

Task:

The task is well designed, although it would have benefited from perhaps one more target speed (for each direction). One monkey appears to have experienced one more target speed than the others (seen in Figure 3C). It would have been nice to have this data for all monkeys, although, of course, unfeasible given that the study has been concluded.

Single unit analyses:

The choice of the three categories (PD shift, gain addition) is not completely justified in a satisfactory way. It would be nice to see whether these three main categories are confirmed by unsupervised methods.

The decoder analyses in Figure 2 provide evidence that target speed modulation may change over the trial. Therefore, it is important to see how the window considered for the firing rate in Figure 1 (currently 100ms pre - 100ms post movement onset) affects the results. Whilst it is of course understandable that a window must be chosen and will always be slightly arbitrary, using different windows and comparing the results of two or three different sizes or timed windows would be more convincing that the results are not dependent on this particular window.

RNN:

Mixed selectivity is not analysed in the RNN, which would help to compare the model to the real data where mixed selectivity is common. The CCA and Procrustes analysis are a good start to validate the claim of similarity between RNN and neural dynamics, rather than allowing comparisons to be dominated by geometric similarities that may be features of the task. However, some of the disparity values for the Procrustes analysis are quite high, albeit below that of the shuffle. Maybe a comment about this in the text should be included. There is also an absence of alternate models to compare the perturbation model results to.

---

## [Referee Report · Reviewer #2 (Public review)]

Summary:

In this manuscript, Zhang et al. examine neural activity in motor cortex as monkeys make reaches in a novel target interception task. Zhang et al. begin by examining the single neuron tuning properties across different moving target conditions, finding several classes of neurons: those that shift their preferred direction, those that change their modulation gain, and those that shift their baseline firing rates. The authors go on to find an interesting, tilted ring structure of the neural population activity, depending on the target speed, and find that (1) the reach direction has consistent positioning around the ring, and (2) the tilt of the ring is highly predictive of the target movement speed. The authors then model the neural activity with a single neuron representational model and a recurrent neural network model, concluding that this population structure requires a mixture of the three types of single neurons described at the beginning of the manuscript.

Strengths:

I find the task the authors present here to be novel and exciting. It slots nicely into an overall trend to break away from a simple reach-to-static-target tasks to better characterize the breadth of how motor cortex generates movements. I also appreciate the movement from single neuron characterization to population activity exploration, which generally serves to anchor the results and make them concrete. Further, the orbital ring structure of population activity is fascinating, and the modeling work at the end serves as a useful baseline control to see how it might arise.

Weaknesses:

While I find the behavioral task presented here to be excitingly novel, I find the presented analyses and results to be far less interesting than they could be. Key to this, I think, is that the authors are examining this task and related neural activity primarily with a single-neuron representational lens. This would be fine as an initial analysis, since the population activity is of course composed of individual neurons, but the field seems to have largely moved towards a more abstract "computation through dynamics" framework that has, in the last several years, provided much more understanding of motor control than the representational framework has. As the manuscript stands now, I'm not entirely sure what interpretation to take away from the representational conclusions the authors made (i.e. the fact that the orbital population geometry arises from a mixture of different tuning types). As such, by the end of the manuscript, I'm not sure I understand any better how motor cortex or its neural geometry might be contributing to the execution of this novel task.

Main Comments:

My main suggestions to the authors revolve around bringing in the computation through a dynamics framework to strengthen their population results. The authors cite the Vyas et al. review paper on the subject, so I believe they are aware of this framework. I have three suggestions for improving or adding to the population results:

(1) Examination of delay period activity: one of the most interesting aspects of the task was the fact that the monkey had a random-length delay period before he could move to intercept the target. Presumably, the monkey had to prepare to intercept at any time between 400 and 800 ms, which means that there may be some interesting preparatory activity dynamics during this period. For example, after 400ms, does the preparatory activity rotate with the target such that once the go cue happens, the correct interception can be executed? There is some analysis of the delay period population activity in the supplement, but it doesn't quite get at the question of how the interception movement is prepared. This is perhaps the most interesting question that can be asked with this experiment, and it's one that I think may be quite novel for the field--it is a shame that it isn't discussed.

(2) Supervised examination of population structure via potent and null spaces: simply examining the first three principal components revealed an orbital structure, with a seemingly conserved motor output space and a dimension orthogonal to it that relates to the visual input. However, the authors don't push this insight any further. One way to do that would be to find the "potent space" of motor cortical activity by regression to the arm movement and examine how the tilted rings look in that space. Presumably, then, the null space should contain information about the target movement. The ring tilt will likely be evident if the authors look at the highest variance neural dimension orthogonal to the potent space (the "null space")--this is akin to PC3 in the current figures, but it would be nice to see what comes out when you look in the data for it.

The authors attempt this sort of analysis in the supplement, alongside their dPCA results, but the results seem misinterpreted. The authors do identify one kind of output-potent space using the reach direction components of dPCA, and the reach directions are indeed aligned here. However, they then go on to interpret the target-velocity space as the output-null space, orthogonal to the potent space. There are two problems with this. (1) The target-velocity space is not necessarily orthogonal to the reach-direction space. This is a key aspect of dPCA--while the individual components within a particular marginalization space are orthogonal, the marginalization spaces themselves are not necessarily orthogonal unless they are forced to be (which the authors don't mention doing). (2) Even if the target-velocity space were orthogonal to the reach-direction space, it would not comprise the whole output-null space--such a null space would also include dimensions of neural population activity that have target-velocity/reach-direction interaction, which the authors show is a major component of neural population variance. Incidentally, the dPCA analysis the authors present shows what I would expect from their unsupervised results, but as it is written, the dPCA results are interpreted in a strange or potentially misleading way.

(3) RNN perturbations: as it's currently written, the RNN modeling has promise, but the perturbations performed don't provide me with much insight. I think this is because the authors are trying to use the RNN to interpret the single neuron tuning, but it's unclear to me what was learned from perturbing the connectivity between what seems to me almost arbitrary groups of neurons. It seems to me that a better perturbation might be to move the neural state before the movement onset to see how it changes the output. For example, the authors could move the neural state from one tilted ring to another to see if the virtual hand then reaches a completely different (yet predictable) target. Moreover, if the authors can more clearly characterize the preparatory movement, perhaps perturbations in the delay period would provide even more insight into how the interception might be prepared.

---

## [Referee Report · Reviewer #3 (Public review)]

Summary:

This experimental study investigates the influence of sensory information on neural population activity in M1 during a delayed reaching task. In the experiment, monkeys are trained to perform a delayed interception reach task, in which the goal is to intercept a potentially moving target.

This paradigm allows the authors to investigate how, given a fixed reach end point (which is assumed to correspond to a fixed motor output), the sensory information regarding the target motion is encoded in neural activity.

At the level of single neurons, the authors find that target motion modulates the activity is three main ways: gain modulation (scaling of the neural activity depending on the target direction), shift (shift of the preferred direction of neurons tuned to reach direction), or addition (offset to the neural activity).

At the level of the neural population, target motion information was largely encoded along the 3rd PC of the neural activity, leading to a tilt of the manifold along which reach direction was encoded that was proportional to target speed. The tilt of the neural manifold was found to be largely driven by the variation of activity of the population of gain modulated neurons.

Finally, the authors study the behaviour of an RNN trained to generate the correct hand velocity given the sensory input and reach direction. The RNN units are found to similarly exhibit mixed selectivity to the sensory information, and the geometry of the « neural population » resembles that observed in the monkeys.

Overall, the experiment is well set up to address the question of how sensory information that is directly relevant to the behaviour but does not lead to a direct change in behavioural output modulates motor cortical activity.

The finding that sensory information modulates the neural activity in M1 during motor preparation and execution is non trivial, given that this modulation of the activity must occur in the nullspace of the movement.

The authors provide analyses at both the single neuron and the population level, leading to a relatively complete characterization of the effect of the target motion on neural activity.

Additionally, they start exploring the link between the population geometry and the mixed selectivity of the single neurons in their RNN model. While they could be extended in future work, the analyses of the RNN provide a good starting point to address how exactly the task setup and constraints on the network shape the single neuron selectivity and the population geometry.

---

## [Author Response]

The following is the authors’ response to the original reviews

**Public Reviews:**

**Reviewer #1 (Public Review):**
Summary:This study addresses the question of how task-relevant sensory information affects activity in the motor cortex. The authors use various approaches to address this question, looking at single units and population activity. They find that there are three subtypes of modulation by sensory information at the single unit level. Population analyses reveal that sensory information affects the neural activity orthogonally to motor output. The authors then compare both single unit and population activity to computational models to investigate how encoding of sensory information at the single unit level is coordinated in a network. They find that an RNN that displays similar orbital dynamics and sensory modulation to the motor cortex also contains nodes that are modulated similarly to the three subtypes identified by the single unit analysis.Strengths:The strengths of this study lie in the population analyses and the approach of comparing single-unit encoding to population dynamics. In particular, the analysis in Figure 3 is very elegant and informative about the effect of sensory information on motor cortical activity.The task is also well designed to suit the questions being asked and well controlled.

We appreciate these kind comments.

It is commendable that the authors compare single units to population modulation. The addition of the RNN model and perturbations strengthen the conclusion that the subtypes of individual units all contribute to the population dynamics. However, the subtypes (PD shift, gain, and addition) are not sufficiently justified. The authors also do not address that single units exhibit mixed modulation, but RNN units are not treated as such.

We’re sorry that we didn’t provide sufficient grounds to introduce the subtypes. We have updated this in the revised manuscript, in Lines 102-104 as:

“We determined these modulations on the basis of the classical cosine tuning model (Georgopoulos et al., 1982) and several previous studies (Bremner and Andersen, 2012; Pesaran et al., 2010; Sergio et al., 2005).”

In our study, we applied the subtype analysis as a criterion to identify the modulation in neuron populations, rather than sorting neurons into exclusively different cell types.

Weaknesses:The main weaknesses of the study lie in the categorization of the single units into PD shift, gain, and addition types. The single units exhibit clear mixed selectivity, as the authors highlight. Therefore, the subsequent analyses looking only at the individual classes in the RNN are a little limited. Another weakness of the paper is that the choice of windows for analyses is not properly justified and the dependence of the results on the time windows chosen for single-unit analyses is not assessed. This is particularly pertinent because tuning curves are known to rotate during movements (Sergio et al. 2005 Journal of Neurophysiology).

In our study, the mixed selectivity or specifically the target-motion modulation on reach- direction tuning is a significant feature of the single neurons. We categorized the neurons into three subclasses, not intending to claim their absolute cell types, but meaning to distinguish target-motion modulation patterns. To further characterize these three patterns, we also investigated their interaction by perturbing connection weights in RNN.

Yes, it’s important to consider the role of rotating tuning curves in neural dynamics during interception. In our case, we observed population neural state with sliding windows, and we focused on the period around movement onset (MO) due to the unexpected ring-like structure and the highest decoding accuracy of transferred decoders (Figure S7C). Then, the single-unit analyses were implemented.

This paper shows sensory information can affect motor cortical activity whilst not affecting motor output. However, it is not the first to do so and fails to cite other papers that have investigated sensory modulation of the motor cortex (Stavinksy et al. 2017 Neuron, Pruszynski et al. 2011 Nature, Omrani et al. 2016 eLife). These studies should be mentioned in the Introduction to capture better the context around the present study. It would also be beneficial to add a discussion of how the results compare to the findings from these other works.

Thanks for the reminder. We’ve introduced these relevant researches in the updated manuscript in Lines 422-426 as:

“To further clarify, the discussing target-motion effect is different from the sensory modulation in action selection (Cisek and Kalaska, 2005), motor planning (Pesaran et al., 2006), visual replay and somatosensory feedback (Pruszynski et al., 2011; Stavisky et al., 2017; Suway and Schwartz, 2019; Tkach et al., 2007), because it occurred around movement onset and in predictive control trial-by-trial.”

This study also uses insights from single-unit analysis to inform mechanistic models of these population dynamics, which is a powerful approach, but is dependent on the validity of the single-cell analysis, which I have expanded on below.I have clarified some of the areas that would benefit from further analysis below:(1) Task:The task is well designed, although it would have benefited from perhaps one more target speed (for each direction). One monkey appears to have experienced one more target speed than the others (seen in Figure 3C). It would have been nice to have this data for all monkeys.

A great suggestion; however, it is hardly feasible as the Utah arrays have already been removed.

(2) Single unit analyses:In some analyses, the effects of target speed look more driven by target movement direction (e.g. Figures 1D and E). To confirm target speed is the main modulator, it would be good to compare how much more variance is explained by models including speed rather than just direction. More target speeds may have been helpful here too.

A nice suggestion. The fitting goodness of the simple model (only movement direction) is much worse than the complex models (including target speed). We’ve updated the results in the revised manuscript in Lines 119-122, as “We found that the adjusted R2 of a full model (0.55 ± 0.24, mean ± sd.) can be higher than that of the PD shift (0.47 ± 0.24), gain (0.46 ± 0.22), additive (0.41 ± 0.26), and simple models (only reach direction, 0.34 ± 0.25) for three monkeys (1162 neurons, ranksum test, one-tailed, p<0.01, Figure S5).”

The choice of the three categories (PD shift, gain addition) is not completely justified in a satisfactory way. It would be nice to see whether these three main categories are confirmed by unsupervised methods.

A good point. It is a pity that we haven’t found an appropriate unsupervised method.

The decoder analyses in Figure 2 provide evidence that target speed modulation may change over the trial. Therefore, it is important to see how the window considered for the firing rate in Figure 1 (currently 100ms pre - 100ms post movement onset) affects the results.

Thanks for the suggestion and close reading. Because the movement onset (MO) is the key time point of this study, we colored this time period in Figure 1 to highlight the perimovement neuronal activity.

(3) Decoder:One feature of the task is that the reach endpoints tile the entire perimeter of the target circle (Figure 1B). However, this feature is not exploited for much of the single-unit analyses. This is most notable in Figure 2, where the use of a SVM limits the decoding to discrete values (the endpoints are divided into 8 categories). Using continuous decoding of hand kinematics would be more appropriate for this task.

This is a very reasonable suggestion. In the revised manuscript, we’ve updated the continuous decoding results with support vector regression (SVR) in Figure S7A and in Lines 170-173 as:

“These results were stable on the data of the other two monkeys and the pseudopopulation of all three monkeys (Figure S6) and reconfirmed by the continuous decoding results with support vector regressions (Figure S7A), suggesting that target motion information existed in M1 throughout almost the entire trial.”

(4) RNN:Mixed selectivity is not analysed in the RNN, which would help to compare the model to the real data where mixed selectivity is common. Furthermore, it would be informative to compare the neural data to the RNN activity using canonical correlation or Procrustes analyses. These would help validate the claim of similarity between RNN and neural dynamics, rather than allowing comparisons to be dominated by geometric similarities that may be features of the task. There is also an absence of alternate models to compare the perturbation model results to.

Thank you for these helpful suggestions. We have performed decoding analysis on RNN units and updated in Figure S12A and Lines 333-334 as: “First, from the decoding result, target motion information existed in nodes’ population dynamics shortly after TO (Figure S12A).”

We also have included the results of canonical correlation analysis and Procrustes analysis in Table S2 and Lines 340-342 as: “We then performed canonical component analysis (CCA) and Procrustes analysis (Table S2; see Methods), the results also indicated the similarity between network dynamics and neural dynamics.”

**Reviewer #2 (Public Review):**
Summary:In this manuscript, Zhang et al. examine neural activity in the motor cortex as monkeys make reaches in a novel target interception task. Zhang et al. begin by examining the single neuron tuning properties across different moving target conditions, finding several classes of neurons: those that shift their preferred direction, those that change their modulation gain, and those that shift their baseline firing rates. The authors go on to find an interesting, tilted ring structure of the neural population activity, depending on the target speed, and find that (1) the reach direction has consistent positioning around the ring, and (2) the tilt of the ring is highly predictive of the target movement speed. The authors then model the neural activity with a single neuron representational model and a recurrent neural network model, concluding that this population structure requires a mixture of the three types of single neurons described at the beginning of the manuscript.Strengths:I find the task the authors present here to be novel and exciting. It slots nicely into an overall trend to break away from a simple reach-to-static-target task to better characterize the breadth of how the motor cortex generates movements. I also appreciate the movement from single neuron characterization to population activity exploration, which generally serves to anchor the results and make them concrete. Further, the orbital ring structure of population activity is fascinating, and the modeling work at the end serves as a useful baseline control to see how it might arise.

Thank you for your recognition of our work.

Weaknesses:While I find the behavioral task presented here to be excitingly novel, I find the presented analyses and results to be far less interesting than they could be. Key to this, I think, is that the authors are examining this task and related neural activity primarily with a singleneuron representational lens. This would be fine as an initial analysis since the population activity is of course composed of individual neurons, but the field seems to have largely moved towards a more abstract "computation through dynamics" framework that has, in the last several years, provided much more understanding of motor control than the representational framework has. As the manuscript stands now, I'm not entirely sure what interpretation to take away from the representational conclusions the authors made (i.e. the fact that the orbital population geometry arises from a mixture of different tuning types). As such, by the end of the manuscript, I'm not sure I understand any better how the motor cortex or its neural geometry might be contributing to the execution of this novel task.

This paper shows the sensory modulation on motor tuning in single units and neural population during motor execution period. It’s a pity that the findings were constrained in certain time windows. We are still working on this task, please look forward to our following work.

Main Comments:My main suggestions to the authors revolve around bringing in the computation through a dynamics framework to strengthen their population results. The authors cite the Vyas et al. review paper on the subject, so I believe they are aware of this framework. I have three suggestions for improving or adding to the population results:(1) Examination of delay period activity: one of the most interesting aspects of the task was the fact that the monkey had a random-length delay period before he could move to intercept the target. Presumably, the monkey had to prepare to intercept at any time between 400 and 800 ms, which means that there may be some interesting preparatory activity dynamics during this period. For example, after 400ms, does the preparatory activity rotate with the target such that once the go cue happens, the correct interception can be executed? There is some analysis of the delay period population activity in the supplement, but it doesn't quite get at the question of how the interception movement is prepared. This is perhaps the most interesting question that can be asked with this experiment, and it's one that I think may be quite novel for the field--it is a shame that it isn't discussed.

It’s a great idea! We are on the way, and it seems promising.

(2) Supervised examination of population structure via potent and null spaces: simply examining the first three principal components revealed an orbital structure, with a seemingly conserved motor output space and a dimension orthogonal to it that relates to the visual input. However, the authors don't push this insight any further. One way to do that would be to find the "potent space" of motor cortical activity by regression to the arm movement and examine how the tilted rings look in that space (this is actually fairly easy to see in the reach direction components of the dPCA plot in the supplement--the rings will be highly aligned in this space). Presumably, then, the null space should contain information about the target movement. dPCA shows that there's not a single dimension that clearly delineates target speed, but the ring tilt is likely evident if the authors look at the highest variance neural dimension orthogonal to the potent space (the "null space")-this is akin to PC3 in the current figures, but it would be nice to see what comes out when you look in the data for it.

Thank you for this nice suggestion. While it was feasible to identify potent subspaces encoding reach direction and null spaces for target-velocity modulation, as suggested by the reviewer, the challenge remained that unsupervised methods were insufficient to isolate a pure target-velocity subspace from numerous possible candidates due to the small variance of target-velocity information. Although dPCA components can be used to construct orthogonal subspaces for individual task variables, we found that the targetvelocity information remained highly entangled with reach-direction representation. More details can be found in Figure S8C and its caption as below:

“We used dPCA components with different features to construct three subspaces (same data in A, reach-direction space #3, #4, #5; target-velocity space #10, #15, #17; interaction space #6, #11, #12), and we projected trial-averaged data into these orthogonal subspaces using different colormaps. This approach allowed us to obtain a “potent subspace” coding reach direction and a “null space” for target velocity. The results showed that the reach-direction subspace effectively represented the reach direction. However, while the target-velocity subspace encoded the target velocity information, it still contained reach-direction clusters within each target-velocity condition, corroborating the results of the addition model in the main text (Figure 4). The interaction subspace revealed that multiple reach-direction rings were nested within each other, similar to the findings from the gain model (Figure 3 & 4). The interaction subspace also captured more variance than target-velocity subspace, consistent with our PCA results, suggesting the target-velocity modulation primarily coexists with reach-direction coding. Furthermore, we explored alternative methods to verify whether orthogonal subspaces could effectively separate the reach direction and target velocity. We could easily identify the reach-direction subspace, but its orthogonal subspace was relatively large, and the target-velocity information exhibited only small variance, making it difficult to isolate a subspace that purely encodes target velocity.”

(3) RNN perturbations: as it's currently written, the RNN modeling has promise, but the perturbations performed don't provide me with much insight. I think this is because the authors are trying to use the RNN to interpret the single neuron tuning, but it's unclear to me what was learned from perturbing the connectivity between what seems to me almost arbitrary groups of neurons (especially considering that 43% of nodes were unclassifiable). It seems to me that a better perturbation might be to move the neural state before the movement onset to see how it changes the output. For example, the authors could move the neural state from one tilted ring to another to see if the virtual hand then reaches a completely different (yet predictable) target. Moreover, if the authors can more clearly characterize the preparatory movement, perhaps perturbations in the delay period would provide even more insight into how the interception might be prepared.

We are sorry that we did not clarify the definition of “none” type, which can be misleading. The 43% unclassifiable nodes include those inactive ones; when only activate (taskrelated) nodes included, the ratio of unclassifiable nodes would be much lower. We recomputed the ratios with only activated units and have updated Table 1. By perturbing the connectivity, we intended to explore the interaction between different modulations.

Thank you for the great advice. We considered moving neural states from one ring to another without changing the directional cluster. However, we found that this perturbation design might not be fully developed: since the top two PCs are highly correlated with movement direction, such a move—similar to exchanging two states within the same cluster but under different target-motion conditions—would presumably not affect the behavior.

**Reviewer #3 (Public Review):**
Summary:This experimental study investigates the influence of sensory information on neural population activity in M1 during a delayed reaching task. In the experiment, monkeys are trained to perform a delayed interception reach task, in which the goal is to intercept a potentially moving target.This paradigm allows the authors to investigate how, given a fixed reach endpoint (which is assumed to correspond to a fixed motor output), the sensory information regarding the target motion is encoded in neural activity.At the level of single neurons, the authors found that target motion modulates the activity in three main ways: gain modulation (scaling of the neural activity depending on the target direction), shift (shift of the preferred direction of neurons tuned to reach direction), or addition (offset to the neural activity).At the level of the neural population, target motion information was largely encoded along the 3rd PC of the neural activity, leading to a tilt of the manifold along which reach direction was encoded that was proportional to the target speed. The tilt of the neural manifold was found to be largely driven by the variation of activity of the population of gain-modulated neurons.Finally, the authors studied the behaviour of an RNN trained to generate the correct hand velocity given the sensory input and reach direction. The RNN units were found to similarly exhibit mixed selectivity to the sensory information, and the geometry of the “ neural population” resembled that observed in the monkeys.Strengths:- The experiment is well set up to address the question of how sensory information that is directly relevant to the behaviour but does not lead to a direct change in behavioural output modulates motor cortical activity.- The finding that sensory information modulates the neural activity in M1 during motor preparation and execution is non trivial, given that this modulation of the activity must occur in the nullspace of the movement.- The paper gives a complete picture of the effect of the target motion on neural activity, by including analyses at the single neuron level as well as at the population level. Additionally, the authors link those two levels of representation by highlighting how gain modulation contributes to shaping the population representation.

Thank you for your recognition.

Weaknesses:- One of the main premises of the paper is the fact that the motor output for a given reach point is preserved across different target motions. However, as the authors briefly mention in the conclusion, they did not record muscle activity during the task, but only hand velocity, making it impossible to directly verify how preserved muscle patterns were across movements. While the authors highlight that they did not see any difference in their results when resampling the data to control for similar hand velocities across conditions, this seems like an important potential caveat of the paper whose implications should be discussed further or highlighted earlier in the paper.

Thanks for the suggestion. We’ve highlighted the resampling results as an important control in the revised manuscript in Figure S11 and Lines 257-260 as:

“To eliminate hand-speed effect, we resampled trials to construct a new dataset with similar distributions of hand speed in each target-motion condition and found similar orbital neural geometry. Moreover, the target-motion gain model provided a better explanation compared to the hand-speed gain model (Figure S11).”

- The main takeaway of the RNN analysis is not fully clear. The authors find that an RNN trained given a sensory input representing a moving target displays modulation to target motion that resembles what is seen in real data. This is interesting, but the authors do not dissect why this representation arises, and how robust it is to various task design choices. For instance, it appears that the network should be able to solve the task using only the motion intention input, which contains the reach endpoint information. If the target motion input is not used for the task, it is not obvious why the RNN units would be modulated by this input (especially as this modulation must lie in the nullspace of the movement hand velocity if the velocity depends only on the reach endpoint). It would thus be important to see alternative models compared to true neural activity, in addition to the model currently included in the paper. Besides, for the model in the paper, it would therefore be interesting to study further how the details of the network setup (eg initial spectral radius of the connectivity, weight regularization, or using only the target position input) affect the modulation by the motion input, as well as the trained population geometry and the relative ratios of modulated cells after training.

Great suggestions. In the revised manuscript, we’ve added the results of three alternative modes in Table S4 and Lines 355-365 as below:

“We also tested three alternative network models: (1) only receives motor intention and a GO-signal; (2) only receives target location and a GO-signal; (3) initialized with sparse connection (sparsity=0.1); the unmentioned settings and training strategies were as the same as those for original models (Table S4; see Methods). The results showed that the three modulations could emerge in these models as well, but with obviously distinctive distributions. In (1), the ring-like structure became overlapped rings parallel to the PC1PC2 plane or barrel-like structure instead; in (2), the target-motion related tilting tendency of the neural states remained, but the projection of the neural states on the PC1-PC2 plane was distorted and the reach-direction clusters dispersed. These implies that both motor intention and target location seem to be needed for the proposed ring-like structure. The initialization of connection weights of the hidden layer can influence the network’s performance and neural state structure, even so, the ring-like structure”

- Additionally, it is unclear what insights are gained from the perturbations to the network connectivity the authors perform, as it is generally expected that modulating the connectivity will degrade task performance and the geometry of the responses. If the authors wish the make claims about the role of the subpopulations, it could be interesting to test whether similar connectivity patterns develop in networks that are not initialized with an all-to-all random connectivity or to use ablation experiments to investigate whether the presence of multiple types of modulations confers any sort of robustness to the network.

Thank you for these great suggestions. By perturbations, we intended to explore the contribution of interaction between certain subpopulations. We’ve included the ablation experiments in the updated manuscript in Table S3 and Lines 344-346 as below: “The ablation experiments showed that losing any kind of modulation nodes would largely deteriorate the performance, and those nodes merely with PD-shift modulation could mostly impact the neural state structure (Table S3).”

- The results suggest that the observed changes in motor cortical activity with target velocity result from M1 activity receiving an input that encodes the velocity information. This also appears to be the assumption in the RNN model. However, even though the input shown to the animal during preparation is indeed a continuously moving target, it appears that the only relevant quantity to the actual movement is the final endpoint of the reach. While this would have to be a function of the target velocity, one could imagine that the computation of where the monkeys should reach might be performed upstream of the motor cortex, in which case the actual target velocity would become irrelevant to the final motor output. This makes the results of the paper very interesting, but it would be nice if the authors could discuss further when one might expect to see modulation by sensory information that does not directly affect motor output in M1, and where those inputs may come from. It may also be interesting to discuss how the findings relate to previous work that has found behaviourally irrelevant information is being filtered out from M1 (for instance, Russo et al, Neuron 2020 found that in monkeys performing a cycling task, context can be decoded from SMA but not from M1, and Wang et al, Nature Communications 2019 found that perceptual information could not be decoded from PMd)?

How and where sensory information modulating M1 are very interesting and open questions. In the revised manuscript, we discuss these in Lines 435-446, as below: “It would be interesting to explore whether other motor areas also allow sensory modulation during flexible interception. The functional differences between M1 and other areas lead to uncertain speculations. Although M1 has pre-movement activity, it is more related to task variables and motor outputs. Recently, a cycling task sets a good example that the supplementary motor area (SMA) encodes context information and the entire movement (Russo et al., 2020), while M1 preferably relates to cycling velocity (Saxena et al., 2022). The dorsal premotor area (PMd) has been reported to capture potential action selection and task probability, while M1 not (Cisek and Kalaska, 2005; Glaser et al., 2018; Wang et al., 2019). If the neural dynamics of other frontal motor areas are revealed, we might be able to tell whether the orbital neural geometry of mixed selectivity is unique in M1, or it is just inherited from upstream areas like PMd. Either outcome would provide us some insights into understanding the interaction between M1 and other frontal motor areas in motor planning.”

**Recommendations for the authors:**

**Reviewer #1 (Recommendations For The Authors):**
At times the writing was a little hard to parse. It could benefit from being fleshed out a bit to link sentences together better.There are a few grammatical errors, such as:"These results support strong and similar roles of gain and additive nodes, but what is even more important is that the three modulations interact each other, so the PD-shift nodes should not be neglected."should be"These results support strong and similar roles of gain and additive nodes, but what is even more important is that the three modulations interact WITH each other, so the PDshift nodes should not be neglected."The discussion could also be more extensive to benefit non-experts in the field.

Thank you. We have proofread and polished the updated manuscript.

**Reviewer #2 (Recommendations For The Authors):**
Other comments:- The authors mention mixed selectivity a few times, but Table 1 doesn't have a column for mixed selective neurons--this seems like an important oversight. Likewise, it would be good to see an example of a "mixed" neuron.- The structure of the writing in the results section often talked about the supplementary results before the main results - this seems backwards. If the supplementary results are important enough to come before the main figures, then they should not be supplementary. Otherwise, if the results are truly supplementary, they should come after the main results are discussed.- Line 305: Authors say "most" RNN units could be classified, and this is technically true, but only barely, according to Table 1. It might be good to put the actual percentage here in the text.- Figure 5a: typo ("Motion intention" rather than "Motor")- I couldn't find any mention of code or data availability in the manuscript.- There were a number of lines that didn't make much sense to me and should probably be rewritten or expanded on:- Lines 167-168: "These results qualitatively imply the interaction as that target speeds..." - Lines 178-179: "However, these neural trajectories were not yet the ideal description, because they were shaped mostly by time."- Lines 187-188: "...suggesting that target motion affects M1 neural dynamics via a topologically invariant transformation."- Lines 224-226: "Note that here we performed an linear transformation on all resulting neural state points to make the ellipse of the static condition orthogonal to the z-axis for better visualization." Does this mean that the z-axis is not PC 3 anymore?- Lines 272-274: "These simulations suggest that the existence of PD-shift and additive modulation would not disrupt the neural geometry that is primarily driven by gain modulation; rather it is possible that these three modulations support each other in a mixed population."

Thank you for these detailed suggestions. By “mixed selectivity”, we mean the joint tuning of both target-motion and movement. In this case, the target-motion modulated neurons (regardless of the modulation type) are of mixed selectivity. The term “motor intention” refers to Mazzoni et al., 1996, Journal of Neurophysiology. We also revised the manuscript for better readership.

We have updated the data and code availability in Data availability as below:

“The example experimental datasets and relevant analysis code have been deposited in Mendeley Data at https://data.mendeley.com/datasets/8gngr6tphf. The RNN relevant code and example model datasets are available at https://github.com/yunchenyc/RNN_ringlike_structure.“

**Reviewer #3 (Recommendations For The Authors):**
Minor typos:Line 153: “there were”Line 301: “network was trained to generate”Line 318: “interact with each other”Suggested reformulations :Line 310 : “tilting angles followed a pattern similar to that seen in the data” Line 187 : the claim of a “topologically invariant transformation” seems strong as the analysis is quite qualitative.Suggested changes to the paper (aside from those mentioned in the main review): It could be nice to show behaviour in a main figure panel early on in the paper. This could help with the task description (as it would directly show how the trials are separated based on endpoint) and could allow for discussing the potential caveats of the assumption that behaviour is preserved.

Thank you. We have corrected these typos and writing problems. As the similar task design has been reported, we finally decided not to provide extra figures or videos. Still, we thank this nice suggestion.